JCB Journal of Cell Biology

# Wbox2: A clathrin terminal domain–derived peptide inhibitor of clathrin-mediated endocytosis

Zhiming Chen, Rosa E. Mino, Marcel Mettlen, Peter Michaely, Madhura Bhave, Dana Kim Reed, and Sandra L. Schmid

**Clathrin-mediated endocytosis (CME) occurs via the formation of clathrin-coated vesicles from clathrin-coated pits (CCPs). Clathrin is recruited to CCPs through interactions between the AP2 complex and its N-terminal domain, which in turn recruits endocytic accessory proteins. Inhibitors of CME that interfere with clathrin function have been described, but their specificity and mechanisms of action are unclear. Here we show that overexpression of the N-terminal domain with (TDD) or without (TD) the distal leg inhibits CME and CCP dynamics by perturbing clathrin interactions with AP2 and SNX9. TDD overexpression does not affect clathrin-independent endocytosis or, surprisingly, AP1-dependent lysosomal trafficking from the Golgi. We designed small membrane–permeant peptides that encode key functional residues within the four known binding sites on the TD. One peptide, Wbox2, encoding residues along the W-box motif binding surface, binds to SNX9 and AP2 and potently and acutely inhibits CME.**

## Introduction

Clathrin-mediated endocytosis (CME) is the predominant route of receptor entry into cells (Mettlen et al., 2018; Schmid and McMahon, 2007). Clathrin triskelia and AP2 complexes are key constituents of the assembled clathrin-coated pits (CCPs; Brodsky et al., 2001). The AP2 complexes are multifunctional heterotetramers that (1) recruit and trigger the assembly of clathrin on the plasma membrane (Cocucci et al., 2012; Edeling et al., 2006; Godlee and Kaksonen, 2013; Kelly et al., 2014; Owen et al., 2000; Shih et al., 1995); (2) recognize cargo receptors, e.g., transferrin receptor (TfnR; Kelly et al., 2008; Mattera et al., 2011; Ohno et al., 1996; Owen and Evans, 1998; Traub and Bonifacino, 2013); and (3) recruit a myriad of endocytic accessory proteins (EAPs; Merrifield and Kaksonen, 2014; Owen et al., 1999; Praefcke et al., 2004; Schmid et al., 2006; Traub et al., 1999). Clathrin triskelions bear three clathrin heavy chains (CHCs), each of which contain a proximal leg that binds clathrin light chains (CLCs), a distal leg, and an N-terminal domain (TD) that binds AP2 and a subset of EAPs (Fig. 1 A; Kirchhausen and Harrison, 1981; Royle, 2006; Ungewickell and Branton, 1981). Two antiparallel proximal and two antiparallel distal legs form a polygonal edge of the clathrin lattice and provide rigidity to the coat (Musacchio et al., 1999). The TD is a β-propeller comprising seven WD40 repeats that generate binding sites for multiple protein interactions identified in vitro (Dell'Angelica, 2001; Lemmon and Traub, 2012). However, TD mutations that perturb these interaction surfaces do not inhibit CME, raising doubts as to their cellular functions (Collette et al., 2009; von Kleist et al.,

2011; Willox and Royle, 2012). Further studies are needed to understand the role of the TD in CME.

Interfering with clathrin function can suppress CME. Initial studies involved overexpression of the "clathrin hub" that retains only proximal legs. The hub inhibits CME by sequestering CLCs (Bennett et al., 2001; Liu et al., 1998); however, inhibition requires high levels of expression (~15-fold) and long incubation times (>20 h) to allow for turnover of endogenous CHCs and sequestration of newly synthesized CLCs. Moreover, hub overexpression causes a dramatic redistribution of endosomes and blocks cargo transport from the Golgi (Bennett et al., 2001; Liu et al., 1998).

A second approach to inhibiting clathrin function uses small-molecule inhibitors called pitstops, which were identified in a screen for chemical inhibitors that block EAP binding to the clathrin-box (Cbox) motif binding site on TD (von Kleist et al., 2011). Like the clathrin hub, in addition to CME, pitstop inhibits AP1-dependent clathrin functions such as recycling of mannose-6-phosphate receptors (M6PRs) from endosomes to the trans-Golgi network (TGN; Stahlschmidt et al., 2014). Moreover, subsequent studies have questioned the specificity and mechanism of action of pitstops (Lemmon and Traub, 2012; Liashkovich et al., 2015; Smith et al., 2013; Willox et al., 2014). For example, pitstop also inhibits clathrin-independent endocytosis (CIE) of multiple cargoes, even in clathrin-depleted cells (Dutta et al., 2012), and inhibits Tfn uptake via CME even in cells expressing Cbox mutants predicted to be defective in EAP interactions at

Department of Cell Biology, University of Texas Southwestern Medical Center, TX.

Correspondence to Sandra L. Schmid: sandra.schmid@utsouthwestern.edu.

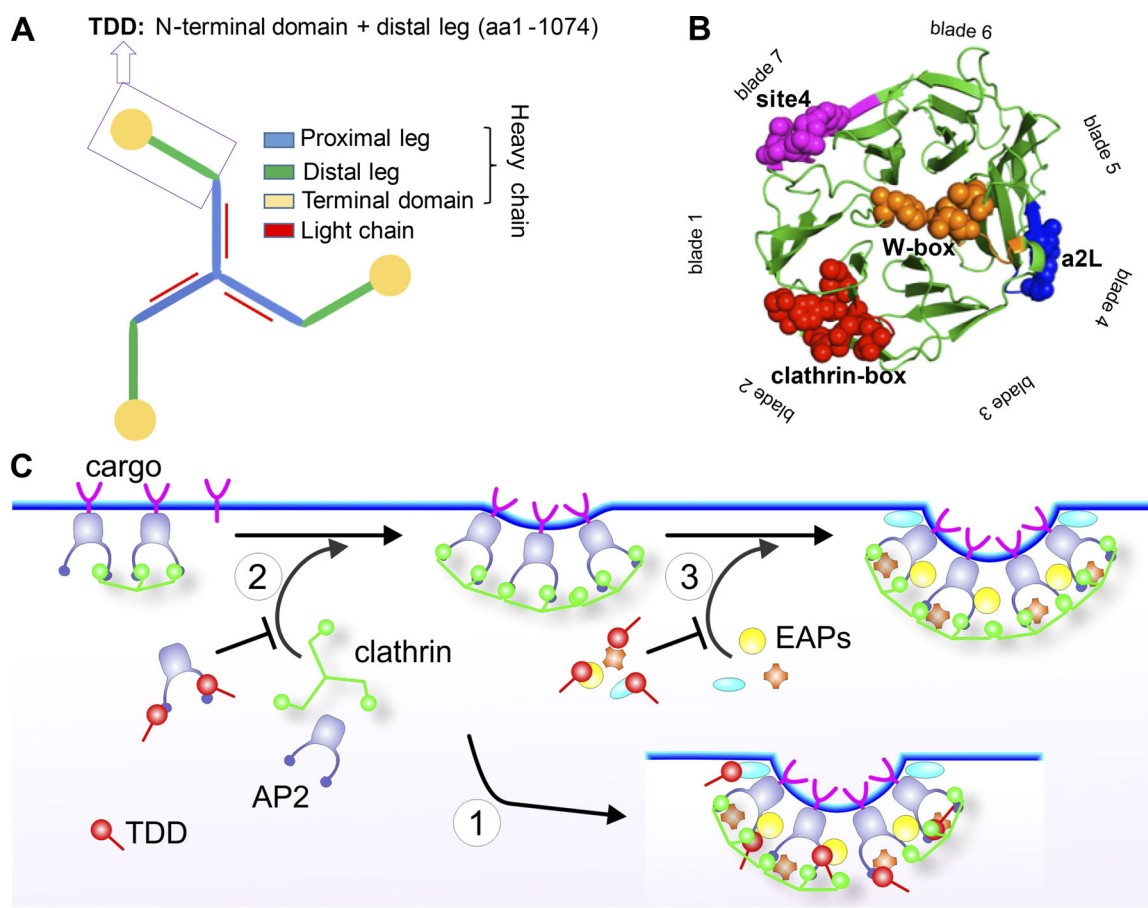

Figure 1. **TDD structure and three possible mechanisms of CME inhibition. (A)** Domain structure of clathrin triskelion. Box indicates the TDD of CHC construct used in this study. **(B)** TD of CHC (PDB accession no. 1BPO). Four reported binding sites are labeled with different colors, and key functional residues are shown as spheres. **(C)** Cartoon to illustrate potential TDD inhibitory mechanisms: (1) TDD is incorporated into and destabilizes/weakens the clathrin coat, thus inhibiting CCP maturation; (2) TDD competes for AP2 and inhibits AP2–clathrin interactions; (3) TDD competes for other EAPs required for CCP growth and maturation.

this site (Lemmon and Traub, 2012; Willox et al., 2014). In addition to the controversy as to the mechanism of action, pitstop is cytotoxic at concentrations not much higher than those required to inhibit CME (Rosselli-Murai et al., 2018), perhaps as a result of its reported effects on spindle assembly (Smith et al., 2013) and nucleocytoplasmic permeability (Liashkovich et al., 2015). Thus, there remains a need for specific inhibitors of CME that work by rapid, well-defined mechanisms.

Here we show that when overexpressed, the clathrin TD potently and specifically inhibits CME by interfering with clathrin–AP2 and clathrin–SNX9 interactions. Moreover, we identify a small membrane–permeant peptide, called Wbox2, that encodes key residues that define the Wbox motif binding site. Wbox2 also binds AP2 and SNX9 and potently, but acutely, inhibits CME, phenocopying the effects of TD overexpression on both early and late stages of CCP maturation.

## Results

To develop a potent inhibitor for CME, we chose a C-terminally truncated fragment of the CHC that encodes the TD, the ankle, and the distal leg (Fig. 1 A). Interestingly, recent genetic studies

have revealed numerous frameshift mutations in the gene encoding CHC linked to neurological diseases that result in the expression of N-terminal fragments corresponding to our TD plus distal leg (TDD) construct (DeMari et al., 2016; Hamdan et al., 2017). We reasoned that this TDD fragment, which encompasses both the TD responsible for interactions with AP2 and EAPs (Lemmon and Traub, 2012; Willox and Royle, 2012; Fig. 1 B), and the distal leg regions of CHC involved in clathrin self-assembly (Musacchio et al., 1999), might inhibit CME by three distinct, but not mutually exclusive, mechanisms. First, because both the distal and proximal legs are key constituents of polygonal edges of the clathrin lattice that contribute to the rigidity of the assembled coat (Greene et al., 2000; Musacchio et al., 1999), overexpressed TDD could be incorporated into and destabilize clathrin-coated structures (CCSs) by displacing intact clathrin triskelia (Fig. 1 C, Mechanism 1). Second, TDD could compete for AP2 binding and inhibit clathrin recruitment and assembly (Fig. 1 C, Mechanism 2). Third, TDD overexpression could compete for interactions between clathrin and other EAPs, thereby inhibiting CCP nucleation, stabilization, and maturation (Fig. 1 C, Mechanism 3). The TDD fragment was cloned into a tetracycline (Tet)-regulated (Tet-off) adenoviral

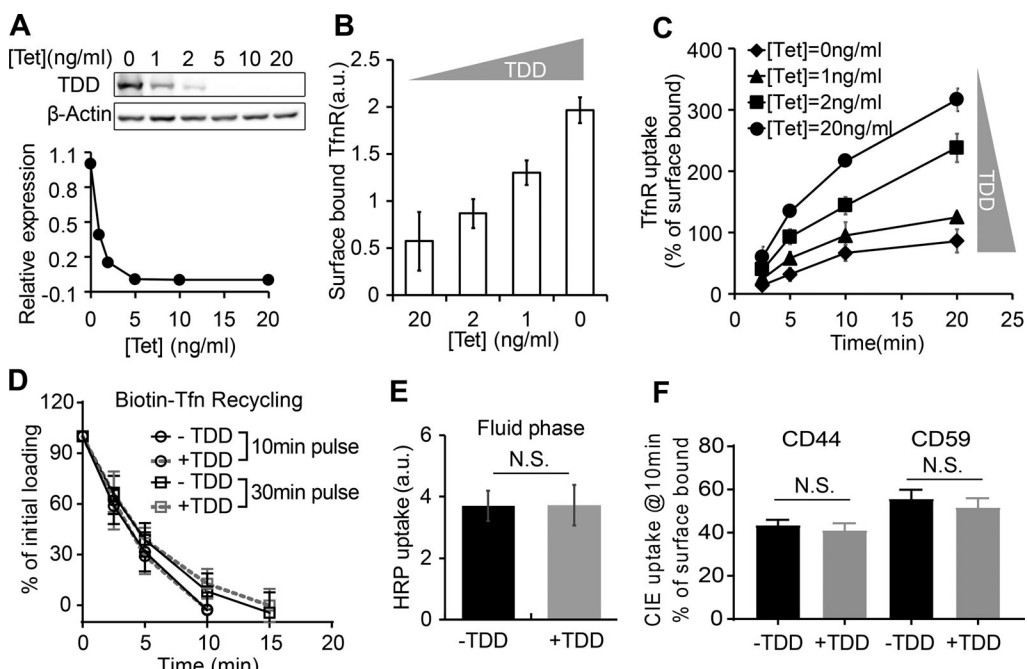

Figure 2. **TDD specifically inhibits CME in a concentration-dependent manner. (A)** Regulation of TDD expression was achieved by adjusting the Tet concentration in the Tet-off system. 350,000 ARPE cells in a six-well plate were infected with 12 µl TDD recombinant adenovirus and 12 µl tTA recombinant adenoviruses in the presence of the indicated concentrations of Tet. **(B and C)** Effects of increasing levels of TDD expression on the levels of surface bound TfnR (B) and the rate and efficiency of TfnR uptake (C) assayed in adenovirus-infected ARPE cells in the presence of the indicated concentrations of Tet. **(D)** Effects of TDD expression (0 ng/ml Tet) on Biotin-Tfn recycling of a 10- or 30-min pulse in adenovirus-infected ARPE/HPV cells with or without Tet. **(E and F)** Effect of TDD expression (0 ng/ml Tet) on the fluid phase uptake of HRP (E) or CIE (F) of CD44 or CD59 in ARPE/HPV cells. [Tet] = 20 ng/ml in D–F. Data ± SD are from $n$ = 4 replicates. N.S., not significant.

system, and cells were coinfected with adenovirus encoding the Tet-regulatable transcription factor tTA. The amounts of TDD and helper Tet-controlled transactivator (tTA) adenoviruses used were optimized for uniform infection of ARPE cells (Fig. S1, A and B), such that the levels of TDD expression could be regulated by adjusting Tet concentration (Fig. 2 A).

## TDD specifically inhibits CME in a concentration-dependent manner

Increasing TDD expression inhibited CME, as evidenced by increased surface levels of TfnR and surface-bound Tfn and by decreased TfnR/Tfn internalization (Fig. 2, B and C; and Fig. S1, C–F). At maximum TDD expression, the endocytic efficiency of TfnR was inhibited by 80% (Fig. 2 C). Recycling of Tfn was not significantly perturbed by TDD expression (Fig. 2 D). Similarly, clathrin-independent endocytic pathways were not significantly inhibited, as measured using either HRP as a marker for fluid phase uptake or CD44 and CD59 as markers for CIE (Fig. 2, E and F). Thus, under these conditions, overexpressed TDD functioned as a potent and specific inhibitor of CME.

## TDD inhibits CCP stabilization and maturation

To define which stages of CME are perturbed by TDD overexpression, we used quantitative live-cell total internal reflection fluorescence microscopy (TIRFM) to visualize and analyze CCP dynamics in ARPE/HPV cells stably expressing eGFP-tagged CLCa (eGFP-CLCa; Mettlen and Danuser, 2014). TDD overexpression

sharply suppressed CCP dynamics. The kymographs of time-lapse imaging showed that large, static clathrin structures predominated when TDD was overexpressed (Fig. 3, A and B; and Videos 1 and 2), indicative of a late block in CCP maturation. FRAP analysis revealed that clathrin turnover in these static structures was also impaired (Fig. S2, A–C). We also noted a dynamic subpopulation of CCPs (arrowheads, Fig. 3 B) which, although visibly obscured by the large static CCPs in kymographs, nonetheless represent the majority of total clathrin-labeled structures detected. Thus, we investigated the effects of TDD on earlier stages of CME using cmeAnalysis (Aguet et al., 2013; Jaqaman et al., 2008; Loerke et al., 2011) to quantify the dynamic behaviors of this subpopulation. When TDD was overexpressed, CCP initiation rates dramatically decreased, with a corresponding increase in dim and transient CCSs (i.e., subthreshold CCSs; Fig. 3 C). The rate and extent of clathrin recruitment (Fig. 3, D and E) also decreased. Together, these changes led to a shift toward dimmer structures (Fig. 3 F). Finally, the remaining dynamic CCPs also exhibited increased lifetimes upon TDD overexpression (Fig. 3, G and H). Thus, we conclude that TDD overexpression inhibits early stages of CCP assembly and stabilization, as well as late stages of CCP maturation.

## TDD is not incorporated into clathrin/AP2 coats but binds to AP2 and SNX9

We next probed the mechanisms by which TDD inhibited CME (Fig. 1 C). TDD overexpression did not alter endogenous

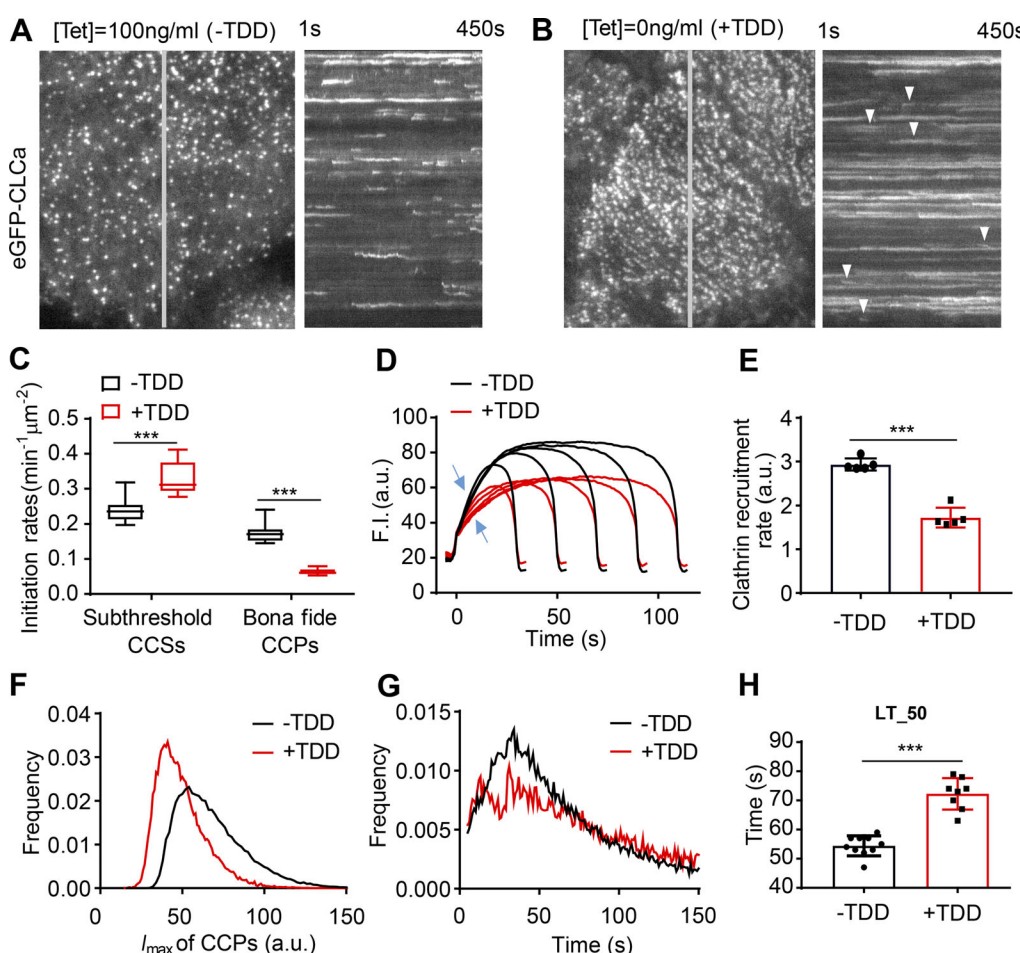

Figure 3. **Effects of TDD expression on CCP dynamics. (A and B)** Single frame from TIRFM videos (7.5 min/video, see Videos 1 and 2) and corresponding kymographs from region indicated by gray line of adenovirally infected ARPE/HPV cells expressing eGFP-CLCa without (A) and with (B) TDD expression. Arrowheads point to examples of dynamic CCSs subjected to cmeAnalysis. **(C)** Effect of TDD expression on the growth and stabilization of CCPs, as measured by the rates of initiation of subthreshold CCSs and bona fide CCPs. n = 10 videos; Student's t test. For box and whisker plots, the boxes encompass the 25th and 75th percentiles, the lines represent the medians, and the whiskers show the minima and maxima data points. **(D and E)** Mean clathrin fluorescence intensity traces in lifetime cohorts of analyzed clathrin structures (D) and corresponding rate of clathrin recruitment determined by the mean slopes of growth phase (from t = 3 s to t = 8 s; E), indicated by blue arrows in D. Each dot represents the mean slope of a cohort. **(F)** Maximum fluorescence intensity ($I_{max}$) distribution of analyzed CCPs. **(G)** Lifetime distribution of bona fide CCPs. **(H)** Median lifetime (LT_50) of analyzed CCPs. Each dot represents a video. Data presented were obtained from a single experiment that is representative of four independent repeats. n = 10 videos for each condition. Number of dynamic tracks analyzed: 124,453 for −TDD and 99,846 for +TDD. Error bars are SD. ***, P ≤ 0.001.

expressions of clathrin or AP2 (Fig. S3 A). Unlike the clathrin hub (Bennett et al., 2001; Liu et al., 1998), TDD was recruited to cell membranes (Fig. 4 A); however, subcellular fractionation (Fig. 4 B and Fig. S3, B and C) and immunofluorescence (Fig. S3 D) established that TDD was not incorporated into clathrin/AP2 coats. These observations ruled out the possibility that TDD might destabilize clathrin coats (Mechanism 1, Fig. 1 C), leaving the possibility that TDD might compete with clathrin for binding to AP2 (Mechanism 2) and/or EAPs (Mechanism 3). In support of an AP2/EAP binding mechanism, the TD alone was sufficient to suppress CCP initiation (Fig. 4 C).

We next performed coimmunoprecipitation (coIP) of HA-tagged TDD from cell lysates (Fig. 4 D) to identify in vivo binding partners. Three independent trials identified SNX9 and the AP2 complex as the principal CME-related proteins that were reproducibly enriched in TDD versus control coIPs (Fig. 4 E

and Table S1). All four subunits of AP2, including both isoforms of the α subunit, were identified with high confidence. Although the degree of enrichment of SNX9 was comparatively low, SNX9 was robustly pulled down based on high sequence coverage (Fig. 4 E). Surprisingly, the AP1 subunits were either detected not at all or at very low abundance compared with AP2 subunits (Table S2). Indeed, TDD overexpression did not alter AP1 localization (Fig. S3 E) or Golgi structure (Fig. S3, F and G). Although the pull-down experiments may have missed weak binding partners, the data suggest that TDD inhibits CME primarily through interference of clathrin interactions with AP2 and SNX9.

### TDD inhibits clathrin–AP2 interactions

To test the effects of TDD overexpression on AP2 function, we tracked the dynamics of AP2 structures by TIRFM in ARPE/HPV

Figure 4. **Defining the mechanism of TDD inhibition. (A)** Subcellular fractionation of ARPE cells expressing TDD into membrane and cytosolic fractions. **(B)** Subcellular fractions to enrich for TX-100–resistant TCVs from ARPE/HPV cells expressing TDD. **(C)** TIRFM imaging and quantification of clathrin dynamics in ARPE/HPV eGFP-CLCa cells indicates that TD phenocopies TDD by reducing the initiation density of bona fide CCPs to the same extent. Error bars are SD. **(D)** Cartoon showing the procedure used for immunoprecipitation of HA-TDD from ARPE cell lysate (with or without HA-TDD expression) using HA antibody and Protein G Sepharose 4 Fast Flow antibody purification resin. Mass spectrometry analysis was applied to the coIP samples for protein ID detection. **(E)** AP2 and SNX9 were identified as the only CME-related protein targets that were reproducibly enriched in the TDD coIPs. Values are average of three independent trials. ***, P ≤ 0.001.

cells expressing α-EGFP-AP2. Similar to clathrin, the predominant phenotype seen in kymographs was a dramatic increase in long-lived static AP2 structures upon TDD overexpression (Fig. 5, A and B; and Videos 3 and 4). AP2 turnover was also impaired in these static structures (Fig. S2, D–F). Unlike clathrin, the background intensity of AP2 on the plasma membrane was enhanced by TDD (Fig. 5, C and D), and we detected a corresponding increase in association of AP2 with the membrane pellet (Fig. 5 E; from 59 ± 14% for control cells to 76 ± 14% for cells with TDD overexpression, n = 3). These data indicate that, consistent with structural studies (Cocucci et al., 2012; Edeling et al., 2006; Godlee and Kaksonen, 2013; Kelly et al., 2014; Owen et al., 2000; Shih et al., 1995), TDD–AP2 interactions can stabilize AP2 complexes on the plasma membrane outside of CCPs.

Dual-color TIRFM imaging and cmeAnalysis, using AP2 (α-eGFP-AP2) as the primary detection and clathrin as secondary (Aguet et al., 2013), revealed that clathrin was depleted from a large portion of dynamic AP2 structures upon TDD overexpression (Fig. 5 F). AP2 structures that were not stabilized at CCPs were short lived (Fig. 5, G and H). Correspondingly, assigning clathrin as primary, the fraction of AP2-deficient CCPs increased, and this subpopulation also was short lived (Fig. 5, I–K). From these data, we conclude that TDD inhibits early stages of CME, in part, by disrupting clathrin–AP2 interactions.

## SNX9 depletion phenocopies the inhibitory effects of TDD overexpression

Given that SNX9 interacts with TDD (Fig. 4 E), we compared the effects of SNX9 knockdown on CCP dynamics with those of TDD overexpression. As was previously reported (Schöneberg et al., 2017), SNX9 knockdown increased the number of long-lived, static CCPs visible in kymographs (Fig. 6 A and Video 5). Analysis of the remaining dynamic structures revealed substantial decreases in the rates of CCP initiation (Fig. 6 B) and clathrin recruitment (Fig. 6, C and D). Dynamic CCPs were also dimmer (i.e., smaller; Fig. 6 E) and, as previously reported (Srinivasan et al., 2018), had prolonged lifetimes (Fig. 6 F). Thus, depletion of SNX9 phenocopied many aspects of the effects of TDD overexpression, consistent with overlapping mechanisms of inhibition of CME.

## Design of inhibitory, membrane-permeant, TD-derived peptides

TDD overexpression potently and selectively inhibited CME; however, the extended incubation times needed to induce protein expression limited its utility. Moreover, we were interested in dissecting which binding sites on the TD contributed to its inhibitory effects. Therefore as potential acute inhibitors for CME, we designed membrane-permeant peptides that encode key functional residues that define the binding sites on TD (Fig. 1 B). These included the Cbox (ter Haar et al., 2000),

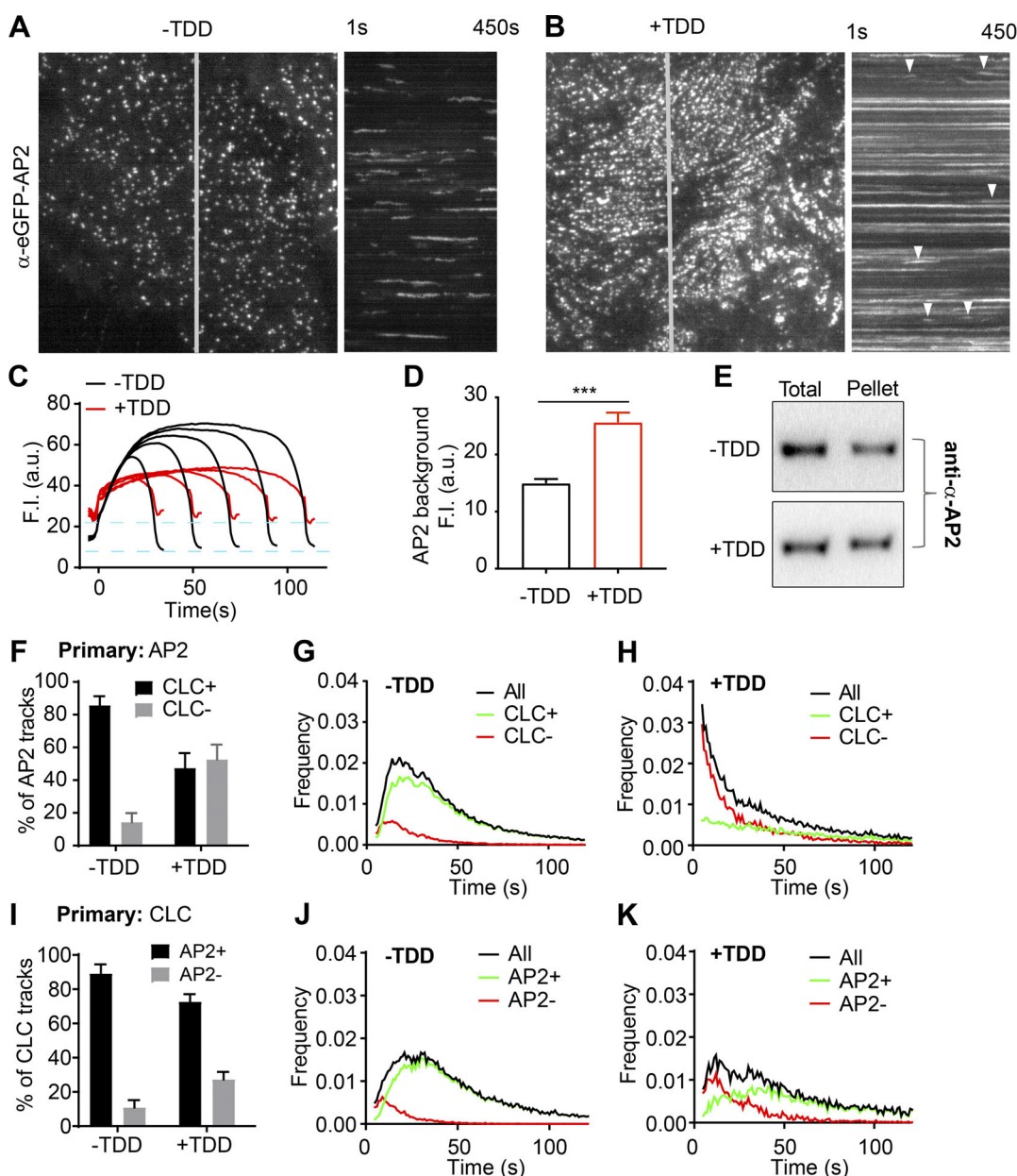

Figure 5. **Effects of TDD on AP2-clathrin interactions. (A–D)** TIRF imaging of ARPE cells expressing α-eGFP-AP2. Data presented were obtained from a single experiment that is representative of three independent repeats. n = 10 videos for each condition. Number of analyzed tracks: 105,498 for −TDD and 102,207 for +TDD. **(A and B)** Single frame from TIRFM video (7.5 min/video, see Videos 3 and 4) and corresponding kymographs without (A) and with (B) TDD expression. Arrowheads point to examples of dynamic structures subjected to cmeAnalysis. **(C)** Mean fluorescence intensity traces of lifetime cohorts of analyzed AP2-labeled structures. Dashed lines indicate the elevation of background fluorescence intensity of AP2 as quantified in D. **(E)** Representative Western blot of α-AP2 from subcellular fractionation of ARPE cells with or without TDD expression. **(F–K)** Dual-color TIRFM imaging of ARPE cells expressing both mRuby-CLCa and α-eGFP-AP2. **(F–H)** Dual-color cmeAnalysis was performed using AP2 as primary channel. Data presented were obtained from a single experiment that is representative of two independent repeats. n = 10 videos for each condition. Number of analyzed tracks: 106,310 for control and 77,932 for +TDD. The percentage of AP2 structures with (CLC+) and without (CLC−) detectable clathrin is shown for −TDD and +TDD cells. **(G and H)** Comparison of lifetime distributions of all detected AP2 structures and those with (CLC+) or without (CLC−) detected clathrin in −TDD (G) and +TDD (H) cells. **(I–K)** Same as F–H, except CLC was assigned as the primary channel. Number of analyzed tracks: 172,166 for −TDD and 133,604 for +TDD. Error bars are SD. ***, P ≤ 0.001.

thought to interact with the classic LLNLD-clathrin binding motif on the β–hinge and other EAPs; the W-box (Miele et al., 2004), thought to interact with tryptophan-containing motifs on SNX9 and amphiphysin; the a2L site, which also binds clathrin binding motif varitants in α-arrestin and β-arrestin (Kang et al., 2009); and the less well-defined site 4 (Willox and Royle, 2012),

also called the "Royle box" (Muenzner et al., 2017; Fig. 1 B). To this end, six peptides encoding the reported key residues within the four binding sites on TD (Fig. 7 A) were designed and synthesized with an N-terminal arginine-rich TAT-sequence for efficient membrane penetration (Copolovici et al., 2014; Walrant et al., 2017; Fig. 7 A). The Cbox1, Wbox1, a2L, and site 4 peptides

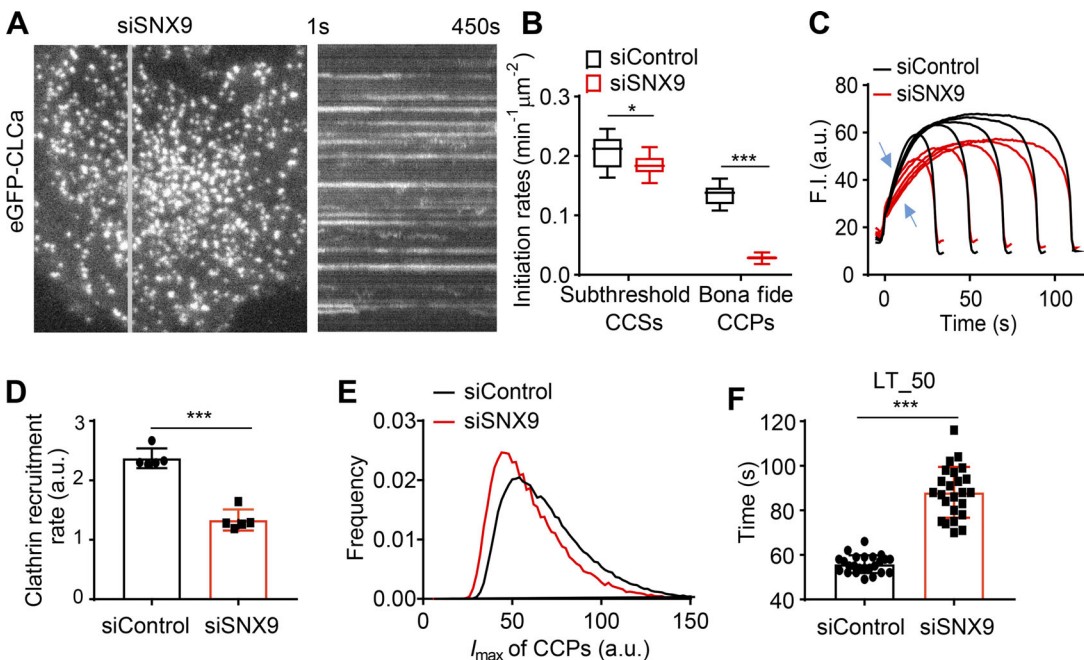

Figure 6. **Effects of SNX9 knockdown on CCP dynamics. (A)** Single frame from TIRFM video (7.5 min/video, see Video 5) and corresponding kymograph from region indicated by the gray line of ARPE/HPV cells expressing eGFP-CLCa. **(B)** Effect of SNX9 knockdown on the initiation rates of bona fide CCPs and subthreshold CCSs. **(C)** Mean clathrin fluorescence intensity traces of lifetime cohorts of analyzed clathrin structures. Growth phase is indicated by blue arrows. **(D)** Mean slopes of the fluorescence intensity cohorts shown in C measured during the growth phase from $t = 3$ s to $t = 8$ s. Each dot represents mean slope of a cohort. **(E)** Maximum fluorescence intensity ($I_{max}$) distribution of CCPs in control and SNX9 knockdown cells. **(F)** Medium lifetime (LT_50) of CCPs in control and SNX9 knockdown cells. Data presented were obtained from a single experiment that is representative of two independent repeats. $n = 24$ videos for each condition. Number of analyzed tracks: 106,310 for control and 77,932 for +TDD. Number of dynamic tracks analyzed: 294,804 for siControl and 152,013 for siSNX9. Error bars are SD. *, $P \leq 0.05$; ***, $P \leq 0.001$.

were derived from linear 10-aa TD sequences (Fig. 7 B). Because key residues in the Cbox and W-box binding sites are encoded on discontinuous residues located on adjacent blades (Fig. 7, A and B), two additional peptides (Cbox2 and Wbox2) were designed to bring these sequences together in a 10-aa peptide engineered to encompass the entire binding motifs.

**Wbox2 is a potent and acute peptide inhibitor of CME**

ARPE/HPV cells were preincubated for 60 min in the presence of 55 µM of each peptide, and their effects on TfnR uptake via CME were measured. Both Wbox and Cbox peptides substantially inhibited CME (Fig. 7 C); the a2L and site4 peptides did not. Similar to TDD, the inhibitory peptides led to the accumulation of TfnRs on the cell surface (Fig. S4 A). Inhibition was rapid and reversible. Using Wbox2 as an example, >50% inhibition was achieved within 10 min of peptide incubation, and CME was largely restored after removing peptides from the culture medium for several hours (Fig. 7, D and E). Concentration curves showed more potent inhibition by Wbox1 and Wbox2 (IC_{50} ~3 µM), compared with Cbox1 (IC_{50} ~60 µM) and Cbox2 (IC_{50} > 100 µM; Fig. 7 F). The TAT peptide, which serves as a negative control, did not inhibit TfnR uptake at concentrations <100 µM. Although their IC_{50}s were comparable, inhibition by Wbox2 exhibited a tighter concentration dependence and achieved a greater degree of inhibition at peptide concentrations >5 µM than that of Wbox1. Mutation of the four reported key residues to alanines greatly reduced the inhibitory activity of the

resulting Wbox2_mutant peptide (Fig. 7 F), consistent with previous studies showing the importance of these residues for protein interactions (Miele et al., 2004). Importantly, Wbox2 had no effect on cell viability at concentrations ≤30 µM and exhibited an IC_{50} for cytotoxicity (>40 µM) that was 10-fold higher than that needed to inhibit CME (Fig. S5).

In addition to the uptake of TfnR and its cargo, Tfn (Fig. S4, B and C), Wbox2 also strongly inhibited the uptake of LDL (Fig. 7 G), even though it can be supported by the Dab2 CME adaptor independently of AP2 (Keyel et al., 2006; Maurer and Cooper, 2006). Moreover, the inhibitory effects of Wbox2 peptides can be extended to other cell types, including normal human fibroblasts, HCC4017 non–small cell lung cancer cells, and HeLa cells (Fig. S4, D–F). The Wbox2 peptide did not significantly perturb Tfn recycling (Fig. S4, G and H), AP1-mediated export of the M6PR from the Golgi (Fig. S4, I–K), or fluid phase uptake (Fig. 7 H). However, unlike TDD, Wbox2 significantly inhibited CD44 and CD59 uptake (Fig. 7 I). Thus, the Wbox2 peptide is a potent, acute, and reversible inhibitor of CME and potentially also of a subset of CIE pathways (see Discussion).

**Wbox2 inhibits CCP dynamics in a manner similar to TDD**

We next investigated the underlying mechanisms of inhibition by Wbox2. Like TDD, application of Wbox2 induced static AP2 and clathrin structures on the plasma membrane (Fig. 8, A and B; and Video 6), indicative of a late block of CCP maturation. The background intensity of AP2 was also elevated by Wbox2

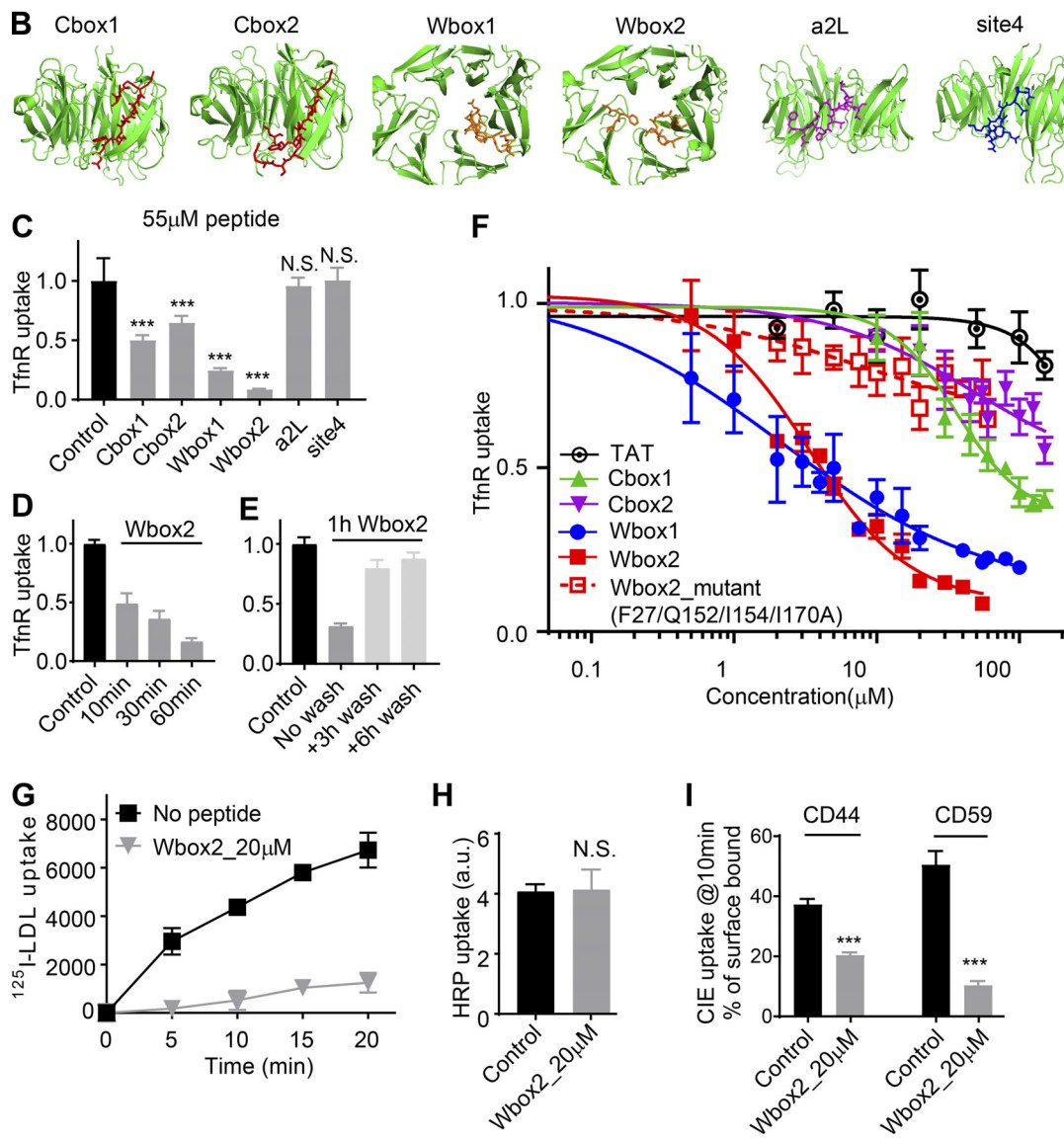

Figure 7. **Peptides designed based on TD binding sites inhibit CME. (A)** Design of TAT-tagged peptides. Previously reported key residues of the four binding sites in clathrin terminal domain are listed on the left column. Peptides encoded the TAT sequence (black) followed by 10 amino acids derived from TD (red, key residues indicated in bold). Cbox1, Wbox1, a2L, and site4 are peptides derived from the linear TD sequences incorporating the key residues. Cbox2 and Wbox2 were designed to incorporate discontinuous key residues located on adjacent blades, bringing them together in a 10-aa peptide engineered to encompass the entire binding motif. **(B)** Amino acid sequences of six designed peptides are shown as colored sticks on the TD (PDB accession no. 1BPO). **(C)** Comparison of the inhibitory activities of the six peptides on TfnR uptake in ARPE/HPV cells. Cells were preincubated with peptides (55 μM) for 60 min at 37°C before uptake assay. Error bars are SD. N.S., not significant. **(D)** Effect of preincubation time (10–60 min at 37°C, as indicated) on the inhibitory activity of 20 μM Wbox2. **(E)** Reversibility of Wbox2 inhibition. TfnR uptake efficiency was measured in ARPE/HPV cells immediately after a 1-h preincubation at 37°C with 20 μM WBox2 or after removing peptides and incubating cells at 37°C for 3 and 6 additional hours. **(F)** Relative TfnR uptake efficiency with increasing concentrations of TAT or the indicated TD-derived peptides, as well as Wbox2_mutant, after 60-min preincubation at 37°C. In C–F, values were normalized to control = 1. Data ± SD are from $n$ = 4 replicates. **(G)** Effect of Wbox2 peptides on $^{125}$I-LDL uptake in human fibroblasts. Data ± SD are from $n$ = 3 replicates. **(H and I)** Effect of Wbox2 peptides (20 μM) on fluid phase uptake of HRP (H) or CIE of CD44 and CD59 (I). Error bars are SD from $n$ = 4 replicates. ***, P ≤ 0.001.

Figure 8. **Wbox2 interferes with AP2–clathrin interactions and alters CCP dynamics.** Dual-color TIRFM imaging of ARPE cells expressing mRuby-CLCa and α-eGFP-AP2 and treated with Wbox2. **(A and B)** Single frame from TIRFM video (7.5 min/video, see Video 6) and corresponding kymographs from region indicated by gray line. **(C)** Background AP2 fluorescence intensity of cells incubated without or with Wbox2 (20 μM). **(D and E)** Dual-color cmeAnalysis with either AP2 (D) or CLC (E) assigned as primary channel showing percentage of α-eGFP-AP2 tracks labeled with mRuby-CLCa (D) or mRuby-CLCa tracks labeled with α-eGFP-AP2 (E) determined in the presence of increasing concentrations of Wbox2. Data presented were obtained from a single experiment that is representative of two independent repeats. Number of videos acquired and analyzed: 11 for 0 μM, 10 for 5 μM, 10 for 10 μM, and 11 for 20 μM. Number of dynamic tracks analyzed in D: 43,324 for 0 μM, 34,664 for 5 μM, 33,596 for 10 μM, and 49,273 for 20 μM. Number of dynamic tracks analyzed in E: 65,192 for 0 μM, 42,323 for 5 μM, 49,733 for 10 μM, and 57,758 for 20 μM. Error bars are SD. ***, $P \le 0.001$.

(Fig. 8 C). As was seen with TDD expression (Fig. 5), Wbox2 decreased the fraction of clathrin-positive AP2 structures (Fig. 8 D), as well as the fraction of AP2-positive clathrin structures (Fig. 8 E), in a concentration-dependent manner. These clathrin-negative AP2 structures and AP2-negative clathrin structures were both short lived (not depicted). These observations motivated us to test whether Wbox2 inhibits early and late stages of CME by interfering with AP2–clathrin and SNX9–clathrin interactions.

**Wbox2 binds SNX9 and AP2 and inhibits AP2–TD interactions**
Isothermal titration calorimetry (ITC) was used to test for and quantitate direct interactions between Wbox2 and purified proteins: SNX9 and the hinge + appendage domain of the AP2 β2 subunit (β2-HAD). Two Wbox2 binding sites were observed for SNX9, with binding affinities of 4.0 and 26.2 μM (Fig. 9 A). Consistent with observations in TfnR uptake assays (Fig. 7 F), TAT and Wbox2_mutant showed much weaker interactions with SNX9 compared with Wbox2. Wbox2 was also observed to bind to a fraction of purified β2-HAD with a binding affinity of 37.5 μM (Fig. 9 A). Given that not all the purified β2-HAD showed active binding with Wbox2, potentially owing to misfolding or interference by the unstructured hinge, we further investigated whether the addition of Wbox2 inhibits interactions of native AP2 in GST-TD (TD residues 1–580) pulldown

experiments. As shown by others (Schmid et al., 2006; Schmid and McMahon, 2007), GST-TD efficiently pulled down AP2 from brain lysates. Incubation in the presence of increasing concentrations of Wbox2 reduced the pulldown efficiency of AP2 by GST-TD at concentrations comparable to its effects on CME (Fig. 9, B and C). As a negative control, TAT and Wbox2_mutant showed much weaker inhibition of TD–AP2 interactions (Fig. 9, B and C). Together, the ability of Wbox2 to interfere with both clathrin–AP2 and clathrin–SNX9 interactions offers a mechanistic basis for its ability to inhibit both early and late stages of CME.

## Discussion
Coated pit assembly begins when AP2 recruits clathrin to the plasma membrane via interactions between the β2-subunit of AP2 and the TD of the CHC (Cocucci et al., 2012; Shih et al., 1995). The seven-bladed β-propeller of TD also binds many EAPs in vitro (Dell'Angelica, 2001; Lemmon and Traub, 2012; ter Haar et al., 2000) and is thus seen as a critical hub in the CME interactome (Schmid et al., 2006; Schmid and McMahon, 2007). TD–hub interactions have been proposed to regulate later stages of CME as the coat matures from an AP2-centric hub to a clathrin-centric hub (Schmid et al., 2006; Schmid and

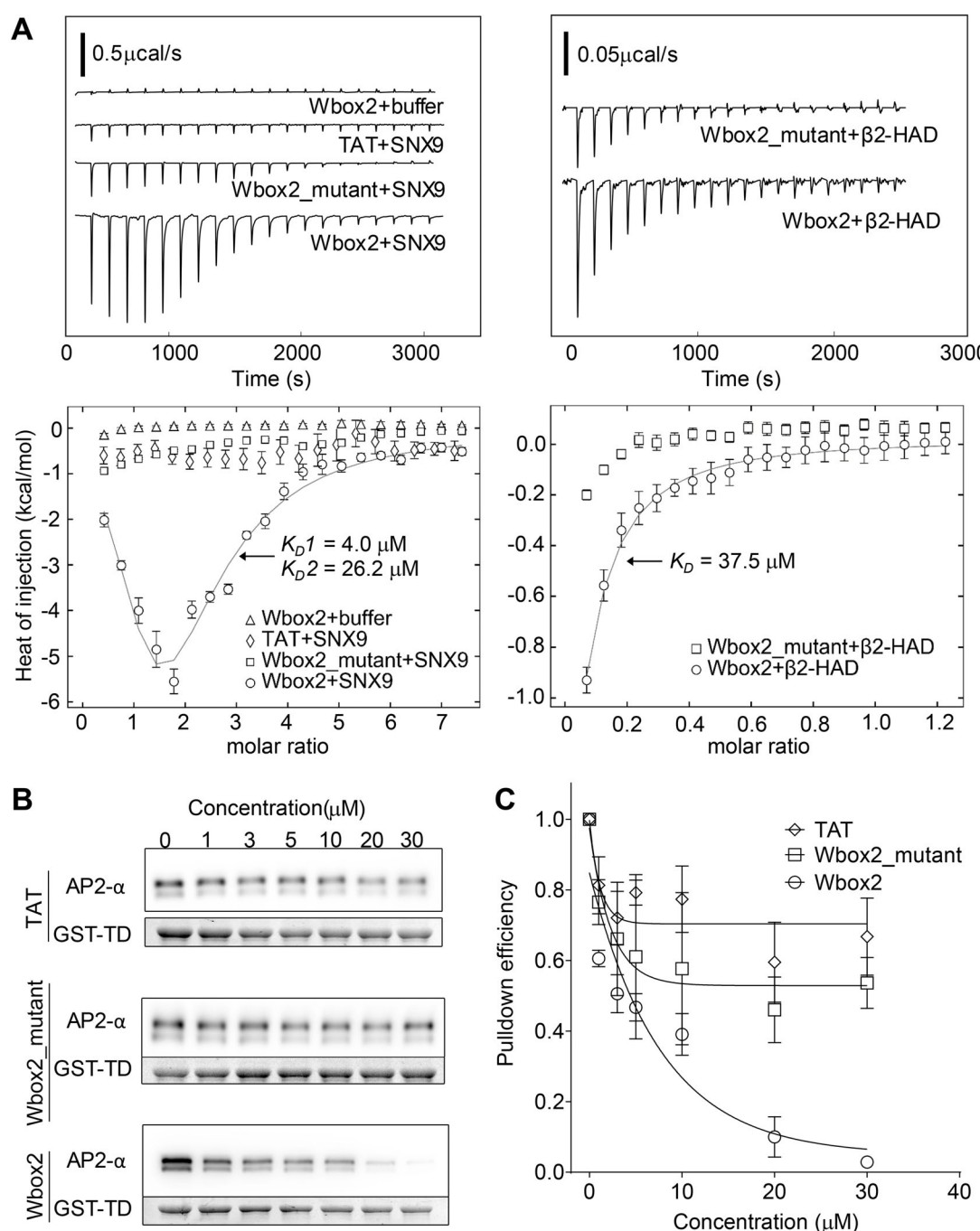

Figure 9. **Wbox2 binds to both AP2 and SNX9. (A)** ITC measurements of peptide binding to SNX9 and β2 hinge + appendage domain (HAD). Heat curves recorded as a function of time during successive 1.9-µl injection of peptides into the cell containing buffer, SNX9, or β2-HAD. Left: Wbox2 + buffer (1.8 mM Wbox2 injected into reaction buffer); TAT + SNX9 (1.8 mM TAT injected into 50 µM SNX9); Wbox2_mutant + SNX9 (1.8 mM Wbox2_mutant injected into 50 µM SNX9); Wbox2 + SNX9 (1.8 mM Wbox2 injected into 50 µM SNX9). Right: Wbox2_mutant + β2-HAD (1.2 mM Wbox2_mutant injected into 200 µM β2-HAD); Wbox2 + β2-HAD (1.2 mM Wbox2 injected into 200 µM β2-HAD). Heat curves were executed with NITPIC and fitted with SEDPHAT. The fitting model for Wbox2 + SNX9 is A+B+B <–> AB+B <–> BAB, with two nonsymmetric sitesK. The fitting model for Wbox2 + β2-HAD is A+B <–> AB, hetero-association. **(B and C)** GST-TD (1–580) pulldown of AP2 from mouse brain extract in the presence of increasing concentrations of TAT, Wbox2_mutant, and Wbox2. Representative Western blot images are presented in B and quantified in C. GST-TD was used as loading control, and each data point represents the average and SD from three independent runs.

McMahon, 2007). However, CME is unaffected by disruption of the major binding sites on the TD, either individually or in combination (Collette et al., 2009; Willox and Royle, 2012). To address this paradox and better define the in vivo function of the TD, we characterized the consequences of its overexpression with (TDD) or without (TD) the distal leg, as a competitive inhibitor of TD interactions. TDD/TD overexpression potently inhibited CME and perturbed CCP dynamics, increasing the

number of static CCPs and lengthening the lifetimes of the dynamic CCPs that remained. Detailed analysis of the dynamic subpopulation of CCPs revealed an increase in transient, dim CCSs but a reduced rate of initiation of bona fide CCPs. This defect in the stabilization of nascent CCPs corresponded to a decrease in the rate and extent of clathrin recruitment. These phenotypes indicate both early (CCP assembly) and late (CCP maturation) roles for TD interactions in CME.

Under our conditions of prolonged (i.e., overnight) overexpression, TDD potently inhibited CME without affecting bulk fluid-phase endocytosis or CIE of CD44 or the GPI-anchored protein CD59. Thus, TDD serves as a new dominant-negative construct able to functionally distinguish uptake by CME and CIE.

Interestingly, recent genetic studies have revealed the existence of de novo frameshift mutations resulting in the heterozygous expression of C-terminally truncated CHC and linked to epilepsy, neurodevelopmental defects and intellectual disabilities (DeMari et al., 2016; Hamdan et al., 2017). As the N-terminal fragments of CHC (TD ± distal leg) correspond to these frameshift mutations, our studies provide insight into the mechanisms contributing to these dominant disease phenotypes.

CoIP identified the AP2 complex and SNX9 as major binding partners of TDD, which together can account for both the early and late effects of TDD overexpression. TDD overexpression reduced the extent of colocalization of AP2 with clathrin and stabilized AP2 on the PM outside of CCPs. Both clathrin-deficient AP2 clusters and AP2-deficient clathrin clusters were short lived, indicating a role for TD–AP2 interactions in stabilizing nascent CCPs. Similarly, SNX9 knockdown phenocopied many of the effects of TDD overexpression, including the formation of static clathrin structures, decreased stabilization of CCPs, slower rates of clathrin recruitment, and prolonged lifetimes of dynamic CCPs. Thus, the inhibitory effects of TDD on CME can be accounted for by its interference with clathrin–AP2 and clathrin–SNX9 interactions (Fig. 1 C, Mechanisms 2 and 3). It was surprising that TDD was not incorporated into CCPs, given that TDD encodes both AP2 binding and clathrin assembly domains. These findings are consistent with the hypothesis that a high degree of cooperativity involving simultaneous interactions with two AP2 complexes is required to initiate clathrin recruitment (Cocucci et al., 2012; Knuehl et al., 2006; Moskowitz et al., 2005).

Our inability to detect interactions between TDD and most EAPs reported to bind TD in vitro, including amphiphysin, eps15, AP180, or epsin1 (Lemmon and Traub, 2012; Royle, 2006; Schmid and McMahon, 2007), is consistent with previous pulldown assays from bovine brain extracts using purified GST-TD (Owen et al., 2000; Schmid et al., 2006). It is possible, as previously suggested (Lemmon and Traub, 2012; Royle, 2006; Schmid and McMahon, 2007), that EAPs bind TDD with low affinity and associate with CCPs only when clathrin assembles into a lattice, creating opportunities for high-avidity interaction.

Unexpectedly, AP1 did not robustly coprecipitate with TDD (Table S2), despite an overall sequence identity of 83% and conservation in the Cbox binding motifs of the β-subunits of AP1, AP2, and AP3 (Ahle and Ungewickell, 1989; Gallusser and Kirchhausen, 1993; ter Haar et al., 2000). Consistent with this,

trafficking from the Golgi, which relies on AP1–clathrin interactions, was not significantly perturbed by TDD overexpression. These findings bring into question the conserved role of Cbox sequences alone in mediating AP–clathrin interactions. Indeed, mutation of the Cbox in the β-subunit of AP3 does not impair AP3 function (Peden et al., 2002). Similarly, a clustered β3 hinge-ear construct is unable to recruit clathrin to the PM, in contrast to an identical β2-hinge-ear construct that was used to "hotwire" CME (Wood et al., 2017). The stronger affinity of AP2 for clathrin may reflect the bipartite nature of AP2–clathrin interactions due to the existence of a second clathrin binding site on the β2-appendage domain (Edeling et al., 2006; Owen et al., 2000). Our findings suggest a need for further exploration of the interactions of AP2, as well as other AP complexes with clathrin.

To overcome the limitations of long-term overexpression of TDD and to probe the contributions of different TD binding sites toward its inhibitory activity, we synthesized peptides encoding the key residues defined by mutagenesis studies that constitute the individual binding surfaces on the TD (Fig. 1; Lemmon and Traub, 2012). Both Wbox- and Cbox-derived peptides significantly inhibited CME. Of these, the Wbox peptides were most potent. Given the β-propeller fold of the TD, the strong inhibitory activity of Wbox2 was somewhat surprising. Several factors can explain these findings. First, the key residues demarking the W-box site are distributed among blades 1, 5, and 6 of the TD β-propeller (Miele et al., 2004). Thus, the β-propeller fold could serve to generate a central interaction surface rather than a binding pocket. Surface binding interactions have the potential of being mimicked by our linear Wbox2 design. Second, the W-box site spans the top and center of the TD (Fig. 1 B) and is thus positioned to take greatest advantage of the bipartite clathrin-interaction surfaces on the β2-hinge and ear (Lemmon and Traub, 2012; Willox and Royle, 2012). Finally, the W-box site faces the membrane in an assembled coat (Fotin et al., 2004) and may thus be better positioned to interact with EAPs involved in cargo recognition or membrane remodeling. Further cell studies would be necessary to test these possibilities. Moreover, we cannot rule out the possibility that modifications of our Cbox peptide design, or of the a2L and site 4 peptides, might improve their efficacy.

Strikingly, treatment of cells with Wbox2 reproduced many of the phenotypes resulting from overexpression of TDD, providing further evidence for the importance of the W-box site in CME. This region of TD is reported to bind proteins via a PWxxW motif, which has been identified only in amphiphysin and SNX9 (Lemmon and Traub, 2012; Miele et al., 2004). We did not detect amphiphysin in any of our pulldown experiments (Table S2); thus, we ascribe some of the effects of Wbox2 to competition of SNX9–TD interactions, as SNX9 plays multiple roles in CME (Schöneberg et al., 2017; Srinivasan et al., 2018). Indeed, Wbox2 binds directly to two sites on SNX9 with affinities comparable to its inhibitory effects on CME. Given that the Cbox site is the presumed binding site for AP2, it was surprising that Wbox2 also phenocopied TD in its ability to interfere with AP2–clathrin interactions. This was confirmed in vitro, as we could detect direct binding of Wbox2 to the β2-hinge-ear and found that it inhibited pulldown of AP2 complexes by

GST-TD, again with affinities comparable to its inhibitory effects on CME.

CHC variants bearing mutations that disrupt the Cbox, the W-box, or both appear to be fully functional in cells (Collette et al., 2009; Willox and Royle, 2012). These observations suggest that AP2 and EAPs can interact with functionally redundant sites on clathrin. Clathrin-binding motifs are promiscuous (Lemmon and Traub, 2012) and often occur in multiple copies within intrinsically disordered regions. The clathrin-binding motifs of intrinsically disordered regions have been proposed to bind to clathrin lattices via a "line-fishing" mechanism involving dynamic and nonspecific association/dissociations without protein folding (Zhuo et al., 2010). The W-box and Cbox peptides could bind to and mask multiple sites on TD-binding partners and thus more globally disrupt these interactions.

That Wbox2 interferes with both AP2– and SNX9–clathrin interactions suggests a mechanistic basis for its ability to phenocopy overexpression of TDD and inhibit both early and late stages of CME. However, unlike TDD, Wbox2 inhibited CD44 and CD59 endocytosis. There are two possibilities for this unexpected difference. The first possibility is that both TDD and WBox2 specifically inhibit CME, but that during prolonged overexpression of TDD, CIE pathways are up-regulated, restoring efficient uptake of CD44 and CD59. In support of this hypothesis, it is likely that cargo such as CD44 and CD59, which have no known interactions with cytoplasmic adaptor proteins, would freely diffuse on the cell surface and be captured, at least in part, by endocytic CCPs. The extent of their uptake via CME- or CIE-dependent pathways might be cell type specific, depending on the relative contributions of CME and CIE to total bulk membrane uptake. For example, Nichols and colleagues (Bitsikas et al., 2014) showed that 90–95% of bulk membrane uptake in unperturbed HeLa, COS, and RPE cells occurred via CME, and that >90% of CD59 was taken up in nascent clathrin-coated vesicles together with Tfn. That CIE pathways are up-regulated following prolonged inhibition of CME, for example by overexpressing dominant-negative dynamins (Bitsikas et al., 2014; Damke et al., 1995) or following knockout of all dynamins (Park et al., 2013), has also been established. Other data consistent with the differential effects of prolonged versus acute inhibition are the findings that CD44 uptake is not inhibited by dominant-negative dynamin overexpression (Mayor et al., 2014) but is inhibited following acute treatment with dynasore (Takahashi et al., 2015). Similarly, acute treatment with pitstop inhibits both CME and CIE (Dutta et al., 2012). The second possibility relates to the ability of Wbox2 to bind two sites on SNX9 with high affinity. Previous studies have shown that siRNA knockdown of SNX9 inhibits CD44 uptake (Bendris et al., 2016) and that SNX9 colocalizes with PM-derived tubules bearing GPI cargo (Yarar et al., 2007). Consistent with this hypothesis is that the Wbox2 inhibits these cargo-selective pathways of CIE while not perturbing bulk fluid phase uptake. Further studies are necessary to distinguish between these two hypotheses, and caution should be taken in interpreting results when using these reagents.

While further studies are underway to design new, more selective peptide-based inhibitors, Wbox2 provides a significant advance over existing chemical inhibitors of CME. Specifically, both dynasore and Dyngo-4a have been shown to have numerous off-target effects unrelated to CME (Park et al., 2013; Persaud et al., 2018). Similarly, pitstop is cytotoxic at concentrations only approximately twofold higher than its reported inhibitory effects on CME (Rosselli-Murai et al., 2018) and has been shown to perturb both spindle pole assembly and nucleo-cytoplasmic permeability (Smith et al., 2013; Liashkovich et al., 2015), although both of these effects have been ascribed to so-called moonlighting functions of clathrin (Brodsky et al., 2014). In contrast, our demonstration that Wbox2 peptide binds AP2 and SNX9 and inhibits AP2–TD interactions at concentrations similar to its inhibitory effects in cells, and that these properties are lost upon mutation of key residues shown to be important for Wbox binding activity, provide a strong basis for the on-target effects of this peptide.

In summary, we show that TDD overexpression inhibited CME primarily through interference of clathrin interactions with SNX9 and the AP2 complex. These interactions were required at multiple stages during CME. We further generated TAT-tagged membrane-permeant peptides based on binding sites on the TD and identified the W-box–derived peptide, Wbox2, that potently and acutely inhibited CME, phenocopying the effects of TDD overexpression. Importantly, we provide insight into the mechanism of Wbox2 inhibition (i.e., its ability to bind AP2 and SNX9), which could account for its early and late effects on CCP maturation, and potentially also for its effects on SNX9-dependent CIE pathways. The design of TAT-tagged peptides able to interfere with specific protein–protein interactions may prove to be a generalizable method for developing acute inhibitors of other cellular processes.

## Materials and methods
### Generation of constructs and viruses
#### TDD constructs
The TDD fragment (residues 1–1,074) in a pET23d vector was a kind gift of Dr. Frances Brodsky (University College London, London, UK), and cloned by seamless technique into a pADT3T7tet vector with an HA tag at the N-terminus. TD was generated by mutating T495 to a stop codon using the TDD construct as template.

#### Adenovirus generation
Recombinant adenoviruses were generated as previously described (Damke et al., 2001; Kadlecova et al., 2017). Briefly, cDNA containing target protein sequence TDD, TD, or tTA was transfected into CRE4 HEK293 cells together with ψ5 DNA. Viruses were harvested by two cycles of freeze–thaw and stored at –80°C. The harvested viruses were used to infect more HEK293 cells for virus expansion.

#### Peptide design, synthesis, and application
Peptides encoding the TAT sequence (YGRKKRRQRRR) followed by 10 amino acid sequences covering the four reported binding sites on the clathrin terminal domain (Fig. 7 A) were synthesized by GenScript (Piscataway, NJ) with >95% purity. Peptides were

dissolved in dH$_2$O and further diluted in PBS4+ (1× PBS buffer plus 0.2% BSA, 1 mM CaCl$_2$, 1 mM MgCl$_2$, and 5 mM D-glucose) or cell culture medium (for imaging) when applied to cells. Peptide solutions were stored at –20°C.

### Cell culture, viral infection, and siRNA knockdown

ARPE19 (herein called ARPE) and ARPE19/HPV16 (herein called APRE/HPV) cells were obtained from ATCC and cultured in DMEM/F12 with 10% FBS. Expression of fluorescent protein-labeling CLCa and/or α subunit of AP2 was achieved by infection with lentiviruses carrying a pLVX-puro vector (mRuby2-CLCa), a pMIB6 vector (eGFP-CLCa), or a modified pMIB6 vector lacking IRIS and BFP expression (α-eGFP-AP2; eGFP is encoded within the linker region of α subunit). Stable cell lines expressing fluorescent tags were sorted by FACS after 72 h. HeLa cells were cultured in DMEM with 10% FBS. Human fibroblast cells were cultured in low-glucose DMEM with 10% FBS. HCC4017 cells were cultured in RPMI 1640 with 5% FBS. All cell lines were cultured at 37°C in 5% CO$_2$.

TDD, TD, and tTA adenoviruses were applied to cells directly in culture medium. Pilot experiments were conducted to optimize the volumes of each virus preparation need for uniform infection (Fig. S1, A and B). Experiments were conducted after 12–18 h, allowing for adequate viral infection and protein expression. Protein expression could be regulated by adjusting Tet concentration. All experiments were performed in the absence of Tet, allowing for maximum protein expression, except for Fig. 1, A–C.

For knockdown of SNX9, transfection of siRNA (Sigma-Aldrich; pool of two siRNAs: UAAGCACUUUGACUGGUUAUU and AACAGUCGUGCUAGUUCCUCA) was conducted in Opti-MEM with Lipofectamine RNAi-MAX (Life Technologies). Cells were plated on a six-well plate (200,000 cells/well) and transfected with siRNA after attaching to the plate. For transfection, 100 µl Opti-MEM with 6.5 µl Lipofectamine RNAi-MAX and 100 µl Opti-MEM with 110 pmol siRNA were incubated at room temperature for 5 min. The two reagents were then mixed and further incubated at room temperature for 10 min. The mixture was added dropwise to cells. Two rounds of transfection were performed through 5 d to achieve >90% target protein knockdown.

### Endocytosis (uptake) assay

Internalization of TfnR, CD44, CD59, and Biotin-Tfn was quantified by in-cell ELISA following established protocols (Conner and Schmid, 2003; Srinivasan et al., 2018). Cells were seeded in 96-well plates (Costar) at 60–70% confluence and grown overnight. Before the assay, cells were starved for 30 min in PBS4+ at 37°C or treated with peptides in PBS4+ at 37°C for the indicated times. After starvation, cells were moved to 4°C, and culture medium was replaced with cold PBS4+ containing 5 µg/ml HTR-D65 (anti-TfnR mAb; Schmid and Smythe, 1991), anti-human CD44 or anti-human CD59 (BD PharMingen) or 5 µg/ml Biotin-Tfn (Sigma-Aldrich). For single-round assays, the above reagents were kept with cells for 20 min at 4°C and then washed out before uptake assay. For multiround assays, the above reagents were kept with cells throughout the uptake process. In

parallel, some cells were kept at 4°C for the measurement of surface-bound ligands and blank controls, and some were incubated in a 37°C water bath for the indicated times. Acid wash (0.2 M acetic acid and 0.2 M NaCl, pH 2.3) was used to remove surface-bound antibodies, followed by washing with cold PBS. All were fixed with 4% PFA (Electron Microscopy Sciences) in PBS for 30 min at 37°C. 0.1% Triton X-100 was added for 6.5 min to permeabilize cells, followed by addition of blocking buffer. HTR-D65–, CD44-, and CD59-treated cells were blocked with Q-PBS (PBS, 2% BSA, 0.1% lysine, and 0.01% saponin, pH 7.4), and Biotin-Tfn–treated cells were blocked with 2% casein. Surface-bound and internalized HTR-D65, CD44, and CD59 were probed with goat anti-mouse antibody conjugated with HRP (Sigma-Aldrich), and Biotin-Tfn was probed with Streptavidin-POD conjugate (Sigma-Aldrich). Color was developed using o-phenylenediamine dihydrochloride (OPD) solution (Sigma-Aldrich), and absorbance was read at 490 nm (Biotek Synergy H1 Hybrid Reader). Cell number variation among wells was accounted for by bicinchoninic acid assay reading at 562 nm.

Fluid-phase uptake assays were conducted using HRP (Sigma-Aldrich) as the readout. Cells were seeded in 96-well plates (Costar) at 60–70% confluence and grown overnight. Before assay, cell culture medium was changed to PBS4+ and cooled down to 4°C. Ice-cold HRP solution (1 mg/ml) was added to cells and moved to a 37°C water bath for 30 min. In parallel, some cells were kept at 4°C as blank control. All the cells were acid washed (0.2 M acetic acid and 0.2 M NaCl, pH 2.3) and lysed with ELISA blocking buffer (1% Triton X-100, 0.1% SDS, 0.2% BSA, 50 mM NaCl, and 1 mM Tris, pH 7.4) for 1 h at 4°C. Color was developed using OPD solution (Sigma-Aldrich), and absorbance was read at 490 nm (Biotek Synergy H1 Hybrid Reader).

$^{125}$I-LDL uptake was measured as previously described (Lombardi et al., 1993; Michaely et al., 2007). Briefly, cells were treated or not with peptide at the indicated concentrations for 30 min in Medium A (DMEM supplemented with 20 mM Hepes, pH 7.5, and 10% lipoprotein-poor serum) in a 37°C/5% CO$_2$ incubator. Cells were then washed with ice-cold Medium B (bicarbonate-free MEM supplemented with 20 mM Hepes, pH 7.5, and 10% lipoprotein-poor serum) and incubated with 20 µg/ml $^{125}$I-LDL with or without the indicated concentration of peptide in Medium B for 1 h at 4°C. Medium was replaced with warm (37°C) medium, and cells were incubated for the indicated times, after which cells were extensively washed with ice-cold PBS and incubated with 1 mg/ml Protease K in Buffer A (1× PBS and 1 mM EDTA) for 1 h at 4°C. The cell suspension was then centrifuged at 5,000 g for 10 min over a cushion of 10% sucrose in PBS. The tubes were frozen in liquid nitrogen, cut to separate the cell pellet (internal) from the solution (surface-bound material released by protease K), and counted on a gamma counter.

### Recycling assay

Biotin-Tfn recycling was conducted according to previous protocols (Chen et al., 2017). Cells were cultured in biotin-free medium for 48 h, seeded in 96-well plates (Costar) at 60–70% confluence, and grown overnight. Before assay, cells were starved for 45 min in PBS4+ at 37°C or treated with peptides in PBS4+ at 37°C for the indicated times. After starvation, cells

were supplied with 100 µl of 5 µg/ml Biotin-Tfn in PBS4+ for 10 or 30 min to load Biotin-Tfn into cells. When loading was complete, cells were cooled down to 4°C and washed with PBS4+ to remove extra Biotin-Tfn. Some cells were kept at 4°C as loading controls. The remaining cells were first treated with avidin and biocytin to mask the surface-bound Biotin-Tfn, and then incubated at 37°C for the indicated times. All the cells were then acid washed and fixed with 4% PFA as above. Blocking steps were the same as for the Biotin-Tfn uptake assay. Data were expressed as percentage of total intracellular ligand remaining relative to the initial load.

## HA-TDD immunoprecipitation

ARPE cells in a 15-cm dish of ∼90% confluence were infected with adenoviruses and incubated for >15 h at 37°C in 5% $CO_2$ to allow for HA-TDD overexpression. Then cells were detached with 5 ml of 50 mM EDTA at 37°C for 10 min and neutralized with 0.2 ml of 1 M $MgCl_2$. After cells were spun down at 500 $g$, 4°C for 3 min, they were washed with 1 ml ice-cold PBS to remove residual EDTA and $MgCl_2$. Cells were then spun down again and resuspended in 1 ml Hepes/KCl buffer (25 mM Hepes, 150 mM KCl, 1 mM $MgCl_2$, 1 mM EGTA, 1× protease inhibitor, and 1× phosphatase inhibitor, pH 7.4) and 0.5% Triton X-100. After 30-min rotation in a cold room and occasional vortexing, the cells were lysed and spun at 500 $g$, 4°C, for 3 min to remove nuclei. Subsequently, 10 µl HA antibody (0.5–0.7 mg/ml; Sigma-Aldrich) was added to cell lysate containing 0.5 g proteins (determined by Bradford assay) and 0.25% Triton X-100 (adjusted by adding an equal volume of Hepes/KCl buffer without Triton X-100). The reaction was allowed to proceed by rotation at 4°C for 2 h and then immunoprecipitated with 60 µl Protein G Sepharose (Sigma-Aldrich). The sediments were washed twice with Hepes/KCl buffer and 0.25% Triton X-100. The final samples were heat denatured and run on an SDS-PAGE gel before being sent to the proteomics core facility for further sample preparation and mass spectrometry analysis. The identified proteins and abundances were output as a spreadsheet for each sample, and Reactome Pathway Database was used for pathway analysis (Table S1 and Table S2).

## Subcellular fractionation

Confluent ARPE or ARPE/HPV cells in a 10-cm dish were detached with 3 ml of 50 mM EDTA at 37°C for 10 min. Extra EDTA after detachment was neutralized with 0.2 ml of 1 M $MgCl_2$. Cells were then spun down at 500 $g$ for 3 min and washed with cold PBS. After another spin to remove PBS, cells were resuspended in 0.5 ml ice-cold lysis buffer (25 mM Hepes, 250 mM sucrose, 1 mM $MgCl_2$, and 2 mM EGTA, pH 7.4). Cells were then lysed by five rounds of freeze–thaw cycle (rapid freezing in liquid nitrogen and slow thawing in room temperature water bath, followed by brief vortex and 10-s sonication). Intact cells and nuclei were removed by centrifugation for 2 min at 500 $g$. 0.1 ml of lysed sample was kept as total control, and the rest (0.4 ml) underwent ultracentrifugation at 4°C for 30 min at 110,000 $g$ to separate membranes (pellet) from cytosol (supernatant). Supernatant was transferred to another low binding Eppendorf, and the pellet was resuspended with the same volume of lysis buffer (0.4 ml). Total, pellet, and supernatant were heat denatured in 1× sample buffer and probed by Western blotting.

## TCV purification

Assembled clathrin/AP2 coats (TCVs) were isolated after cell lysis in 0.5% Triton X-100, as previously described (Pearse, 1982). Briefly, confluent ARPE/HPV eGFP-CLCa cells in five 15-cm dishes were detached with 50 mM EDTA (5 ml each dish) at 37°C for 5–10 min. Subsequently, extra EDTA was neutralized with 1 ml of 1 M $MgCl_2$. Cells were then collected at 500 $g$ for 3 min and washed with cold PBS. PBS removal was followed by resuspension in 2 ml ice-cold lysis buffer (100 mM Mes, 0.2 mM EGTA, 0.5 mM $MgCl_2$, and 0.5% Triton X-100, pH 6.2). The resuspended cell lysates were split into two low binding Eppendorf tubes and rotated at 4°C for 30 min with occasional vortexing. Lysed cells were spun at 500 $g$ and 4°C for 3 min to remove nuclei before ultracentrifugation at 121,000 $g$ and 4°C for 45 min to collect enriched TCVs in the pellet.

## Immunofluorescence

Cells seeded on gelatin-coated 22 × 22–mm glass (Corning; 2850-22) were rinsed three times with 2 ml PBS and then fixed with 4% PFA (Electron Microscopy Sciences) for 30 min at 37°C, and 0.5% Triton X-100 was applied to permeabilize the cells. After blocking with Q-PBS, primary antibody and secondary antibodies were added consecutively to cells and incubated for 1 h at room temperature with thorough washes in between. Cells were mounted on cover slides with spacers and sealed with PBS for TIRFM or wide-field imaging.

## TIRFM live-cell imaging and CCP quantification of dynamics

Cells were seeded on a gelatin-coated 22 × 22–mm cover glass (Corning; 2850-22) overnight and moved to fresh cell culture medium 30 min before being mounted on a 25 × 75–mm slide (Thermo Fisher Scientific; 3050). Imaging was conducted with a 60×, 1.49-NA Apo TIRF objective (Nikon) mounted on a Ti-Eclipse inverted microscope. Perfect focus was used during time-lapse imaging. TIRF penetration depth was ∼80 nm. Videos were acquired for 7.5 min at the rate of 1 frame/s for all live-cell imaging. For dual-color imaging, two sequential images of primary and secondary channel were taken within 1 s, and the video length was the same as single-channel imaging. Published cmeAnalysis software was used for CCP detection, tracking, and quantification (Aguet et al., 2013; Jaqaman et al., 2008; Loerke et al., 2011). More than 10 cells were imaged per condition, and the number of total analyzed tracks is indicated in the figure legends.

## TEM imaging

Cells grown on gelatin-coated glass-bottomed MatTek dishes were washed with cold PBS and fixed in 2.5% (vol/vol) glutaraldehyde in 0.1 M sodium cacodylate buffer. Processing for embedding and sectioning continued as follows: after three rinses in 0.1 M sodium cacodylate buffer, cells were postfixed in 1% osmium in 0.1 M sodium cacodylate buffer for 1 h. Cells were rinsed in 0.1 M sodium cacodylate buffer and stained en bloc with 0.5% tannic acid in 0.05 M sodium cacodylate buffer for

30 min. After two rinses in 1% sodium sulfate in 0.1 M sodium cacodylate buffer, samples were rinsed three times in 0.1 M sodium cacodylate buffer and five times in water. Samples were then dehydrated through a series of increasing concentrations of ethanol and infiltrated and embedded in Embed-812 resin. Enough resin was added into the MatTek dishes to just fill the well and polymerized at 60°C. Polymerized samples were dropped into liquid nitrogen to pop out the resin disks from the center of the MatTek dish. Two resin disks containing the same sample were sandwiched together with fresh Embed-812, monolayers facing each other. Resin disks were polymerized at 60°C overnight and sectioned along the longitudinal axis of the two monolayers of cells with a diamond knife (Diatome) on an Ultracut UCT 6 ultramicrotome (Leica Microsystems). Sections were poststained with 2% (wt/vol) uranyl acetate in water and lead citrate. Imaging was done on a JEM-1400 Plus transmission electron microscope equipped with a LaB6 source operated at 120 kV using an AMT-BioSprint 16M charge-coupled device camera. Purified TCVs were applied to 200-mesh copper grids and allowed to adsorb for 10 min, after which the grids were dipped in water droplets to wash away unadsorbed material. The grids were then incubated in 2% uranyl acetate for 2 min. After three dips in water droplets, grids were dried and imaged.

### Protein purification and ITC measurements

GST-SNX9 in a pGEX-KG vector and His6x-β2-HAD in a pET vector (gift of Dr. Linton Traub, University of Pittsburgh, Pittsburgh, PA) were expressed in BL21(DE3) and then affinity purified using glutathione Sepharose 4B beads (ABT) and a HisTrap HP column (GE Healthcare). The affinity-purified GST-SNX9 was thrombin-digested to remove GST. The resulting SNX9 and β2-HAD proteins were separately applied to a HiLoad 26/600 Superdex 200 pg column (GE Healthcare). Peak fractions of target proteins were collected and concentrated using Amicon Ultra-15 10K Centrifugal filters (Sigma-Aldrich).

Binding of TAT, Wbox2_mutant, and Wbox2 to SNX9 and β2-HAD was investigated by ITC using a MicroCal iTC$_{200}$ (GE Healthcare Life Sciences). Measurements were performed at 20°C in reaction buffer: 20 mM Hepes, 150 mM NaCl, and 1 mM TCEP, pH 7.4. Protein and peptide concentrations were determined by absorbance at 280 and 205 nm, respectively. The peptides were injected from a syringe in 21 steps up to a molar excess over the protein concentration. Titration curves were executed with NITPIC and fitted with SEDPHAT. GUSSI was further used to output the figures presented.

### GST-TD pulldown from mouse brain extract

A mouse brain was homogenized using a Dounce tissue grinder (Sartorius) in 1 ml MES buffer (100 mM Mes, 1 mM EGTA, 0.5 mM MgCl$_2$, and 0.1 mM PMSF, pH 6.4). After spinning at 5,000 $g$ and 4°C for 3 min, the supernatant was ultracentrifuged at 110,000 $g$ for 60 min. The resulting pellet was then resuspended in extraction buffer (3 vol of 1 M Tris, pH 8, 1 vol Mes buffer, with added PMSF and DTT to 0.1 mM each) for 30 min at room temperature. The resuspended solution was ultracentrifuged at 110,000 $g$ for 60 min to collect supernatant containing released coat proteins. The brain extract buffer was

exchanged to pulldown reaction buffer (50 mM Tris, 150 mM NaCl, and 0.1 mM EGTA, pH 7.5) using zeba spin desalting columns (Thermo Fisher Scientific).

In the pulldown assay, 75 µg GST-TD (residues 1–580) in reaction buffer was mixed with coat protein fractions (10% of total extract from a mouse brain) and the indicated concentrations of peptides. The mixture was incubated with 50 µl of a 50% slurry of glutathione Sepharose 4B beads (ABT) and left to rotate overnight at 4°C. Subsequently, beads were collected by spinning down at 1,000 $g$ for 2 min and washed twice with reaction buffer. The beads were resuspended in Laemmli sample buffer and denatured at 95°C before running on 7.5% SDS-PAGE and Western blotting.

### Quantification and statistical analysis

Endocytosis and recycling assays were performed in biological replicates. TIRFM data were biologically reproduced, and a representative dataset from the same day is presented. The intensity of protein blots was analyzed with ImageJ (National Institutes of Health). For all data, error bars are SD, and statistical significance was analyzed by two-tailed Student's $t$ test. *, P ≤ 0.05; **, P ≤ 0.01; ***, P ≤ 0.001.

### Data availability

Mass spectrometry data that support the findings of this study are available in Table S1 and Table S2. All other data supporting the findings of this study are available from the corresponding author on reasonable request.

### Code availability

The software and algorithms for data analyses used in this study are all well established from previous work and are referenced throughout the manuscript. No custom code was used in this study.

### Reagents

A list of reagents used in this study is supplied as Table S3.

### Online supplemental material

Fig. S1 shows the optimization of recombinant adeno-viral system in ARPE cells (related to Figs. 1 and 2). Fig. S2 is a FRAP experiment showing that TDD overexpression inhibits clathrin and AP2 exchange on the plasma membrane (related to Figs. 3 and 5). Fig. S3 shows the effect of TDD overexpression on endogenous protein expression, clathrin-coat stability, and AP1-mediated clathrin trafficking from Golgi (related to Fig. 4). Fig. S4 shows the effect of TD-derived peptides on CME, Tfn recycling, and AP1-mediated Golgi trafficking (related to Fig. 7). Fig. S5 shows cell viability after Wbox2 treatment (related to Fig. 7). Table S1 contains the list of proteins involved in CME pathway identified by mass spectrometry of HA-TDD pulldowns from three independent trials (related to Fig. 4). Table S2 contains the list of proteins involved in membrane trafficking pathways from three independent mass spectrometry experiments (related to Fig. 4). Table S3 contains the list of reagents used in this work, including chemicals, proteins, antibodies, and siRNAs. Videos 1 and 2 shows time-lapse TIRFM imaging of

stable ARPE/HPV cells expressing eGFP-CLCa without and with TDD overexpression, respectively (related to Fig. 3). Videos 3 and 4 shows time-lapse TIRFM imaging of stable ARPE cells expressing α-eGFP-AP2 without and with TDD overexpression, respectively (related to Fig. 5). Video 5 shows time-lapse TIRFM imaging of stable ARPE/HPV cells expressing eGFP-CLC 5 d after siRNA-mediated knockdown of SNX9 (related to Fig. 6). Video 6 shows time-lapse TIRFM imaging of stable ARPE cells expressing α-eGFP-AP2 and mRuby-CLCa incubated in the presence of 20 µM Wbox2 (related to Fig. 8).

## Acknowledgments

We thank Dr. Frances Brodsky for reagents. We thank the UTSW proteomics core facility for help with sample processing and analysis, the Molecular Biophysics Core facility for the help with ITC, and the Electron Microscopy Core facility for expert assistance in processing EM samples. We acknowledge Aparna Mohanakrishnan for the technical support in PCR mutagenesis and Heather Grossman for the technical support in adenovirus production.

This work is supported by Welch Foundation grant I-1823 and National Institutes of Health grants MH61345, GM73165, and GM42455 to S.L. Schmid and National Institutes of Health grant 1S10OD021685 to the EM Core facility.

The authors declare no competing financial interests.

Author contributions: Z. Chen and S.L. Schmid designed the experiments, interpreted the results, and wrote the manuscript with input from all authors. Z. Chen and M. Mettlen performed the TIRF and wide-field microscope imaging and data analysis. Z Chen R.E. Mino, and M. Bhave performed the TfnR/Tfn uptake and recycling assays. Z. Chen performed the ITC, Cell Counting Kit 8, and pulldown assays. P. Michaely performed the [125]I-LDL uptake assay. D.K. Reed performed the TEM imaging.

Submitted: 23 August 2019

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

# Supplemental material

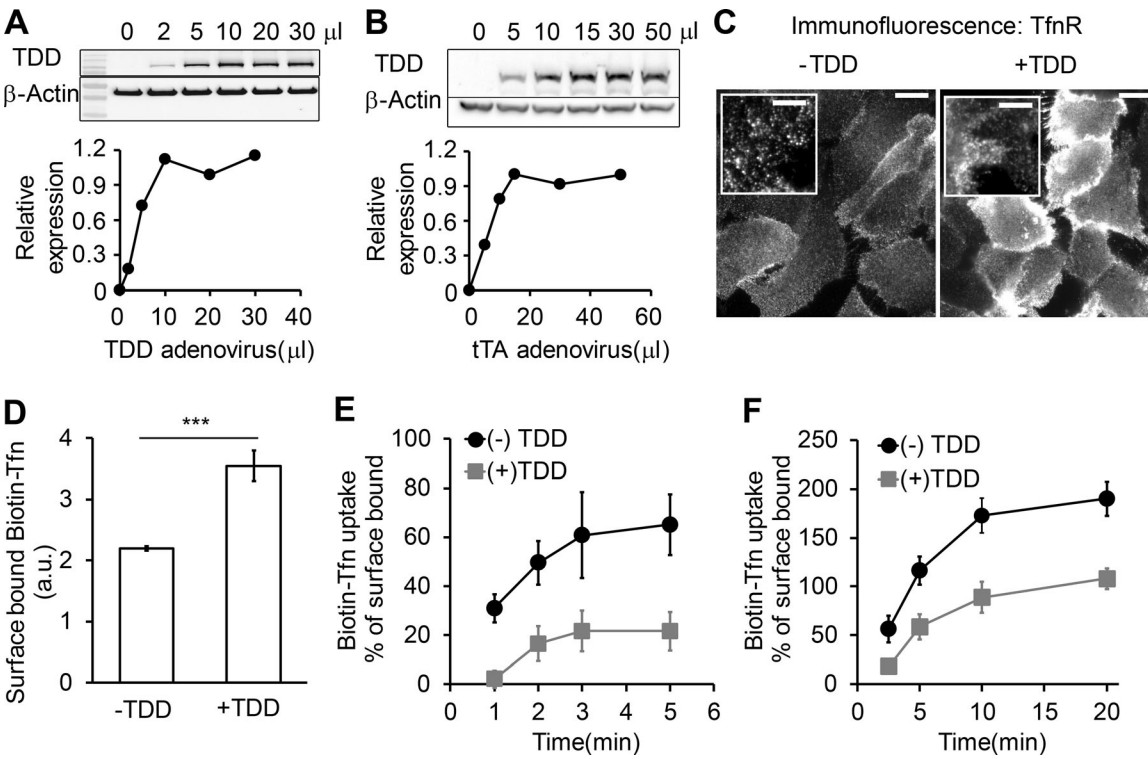

Figure S1. **Optimization of recombinant adenoviral system in ARPE cells.** The amounts of TDD and helper tTA adenoviruses were optimized for uniform infection of ARPE cells. **(A)** Relative TDD expression level after infection with 12 µl tTA recombinant adenovirus and the indicated volumes of TDD recombinant adenoviruses. **(B)** Relative TDD expression level after infection with 20 µl TDD recombinant adenovirus and the indicated volumes of tTA recombinant adenoviruses. **(C)** Immunofluorescence of surface-bound TfnR with or without TDD expression. Scale bars = 20 µm; inset scale bars = 5 µm. **(D–F)** Effects of TDD overexpression on Biotin-Tfn uptake revealed that TDD expression enhanced surface bound Biotin-Tfn (D) and reduced its uptake efficiency in single-round (E) and multiround (F) Biotin-Tfn uptake. Average value ± SD are from *n* = 4 replicates. ***, P ≤ 0.001.

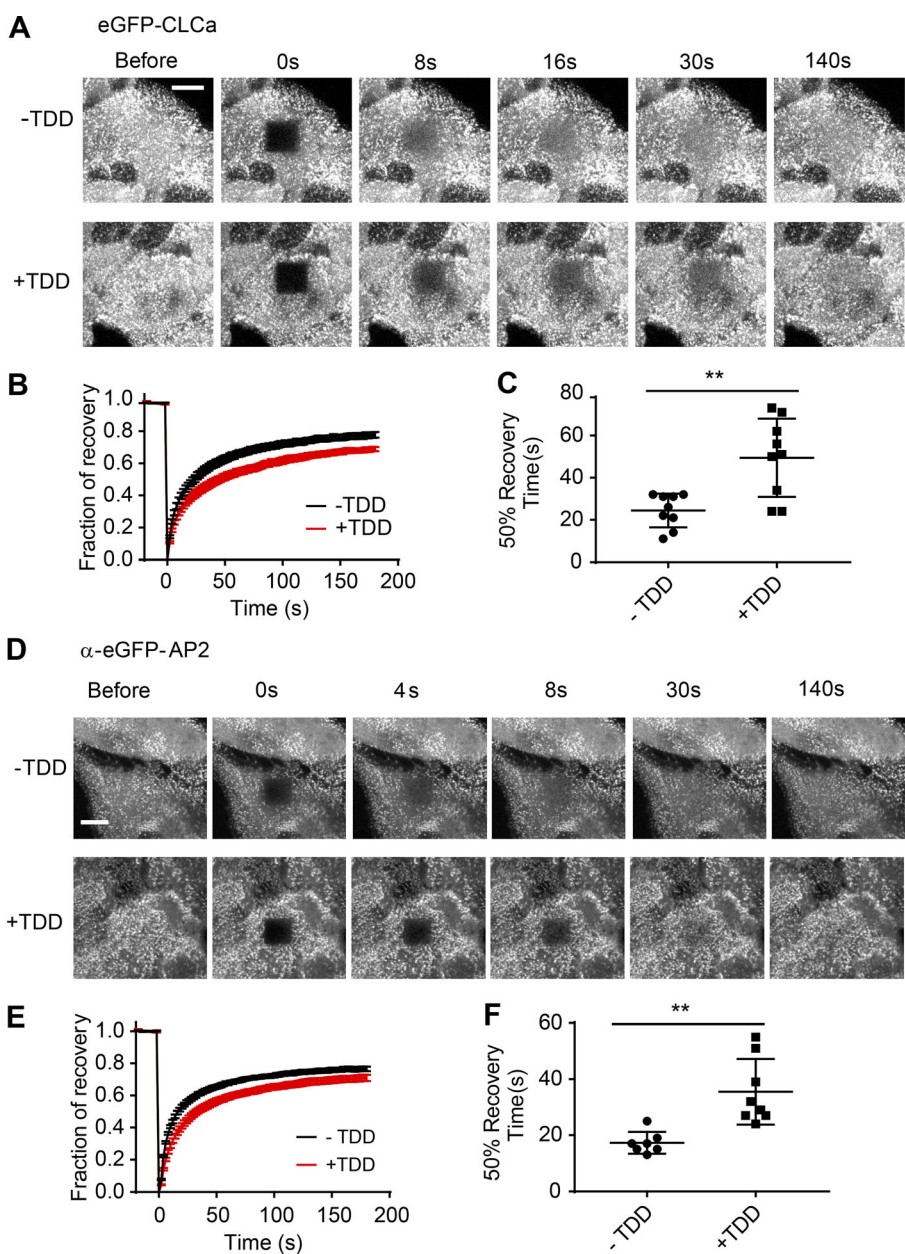

Figure S2. **Clathrin and AP2 exchange on plasma membrane is inhibited by TDD expression**. **(A–C)** FRAP was conducted at 37°C in ARPE/HPV eGFP-CLCa cells with or without TDD overexpression using a confocal microscope. **(A)** Representative time-lapse FRAP images. **(B)** Average fluorescent intensity traces of the photobleached area (dark square at $t$ = 0 s). **(C)** Time required for 50% fluorescence recovery. **(D–F)** FRAP was conducted at 37°C in ARPE α-eGFP-AP2 cells with or without TDD overexpression using a confocal microscope. **(D)** Representative time-lapse FRAP images. **(E)** Average fluorescent intensity traces of the photobleached area (dark square at $t$ = 0 s). **(F)** Time required for 50% fluorescence recovery. Error bars are SD from $n$ = 8 experiments. Scale bars = 3 μm. The lines in these scatter dot plots represent the means with SD. **, P ≤ 0.01.

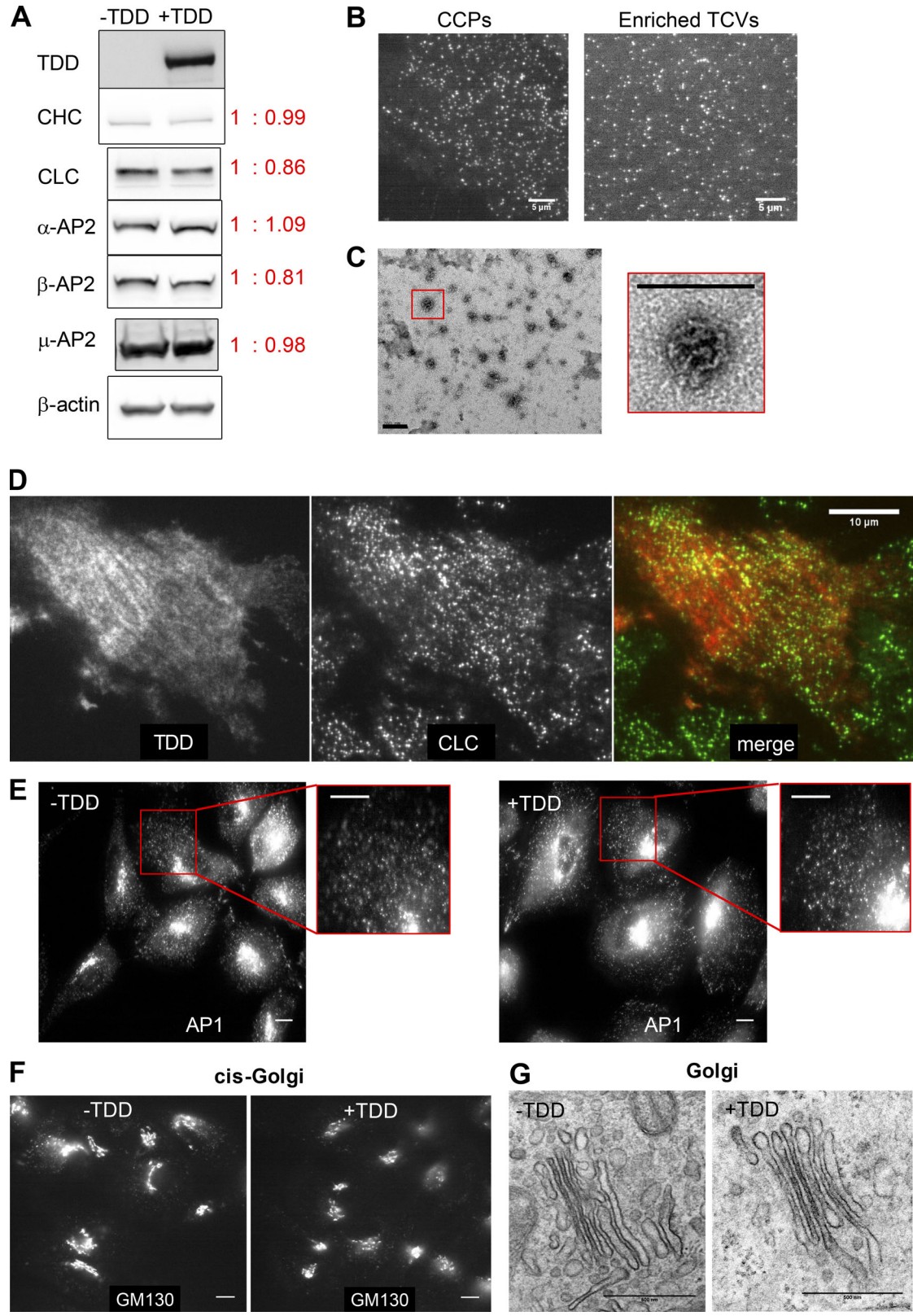

Figure S3. **Effect of TDD expression on endogenous protein expression, clathrin-coat stability, and AP1-mediated clathrin trafficking from Golgi.**
**(A)** ARPE/HPV cells expressing eGFP-CLCa cells were treated with or without TDD overexpression. TDD expression did not alter the expression levels of endogenous clathrin or AP2. **(B)** Representative TIRF images of CCPs in live cells or isolated TCVs. Scale bars = 5 µm. **(C)** Representative EM images showing the collapsed coats typical of isolated TCVs. Scale bars = 200 nm. **(D)** Immunostaining of HA-TDD in ARPE/HPV eGFP-CLCa cells. Scale bar = 10 µm. **(E–G)** ARPE cells were treated with or without TDD overexpression. Immunostaining of AP1 γ subunit (E) and GM-130 cis-Golgi marker (F) were imaged with wide-field microscopy. Scale bars = 10 µm. **(G)** Representative EM images of Golgi apparatus. Scale bars = 500 nm.

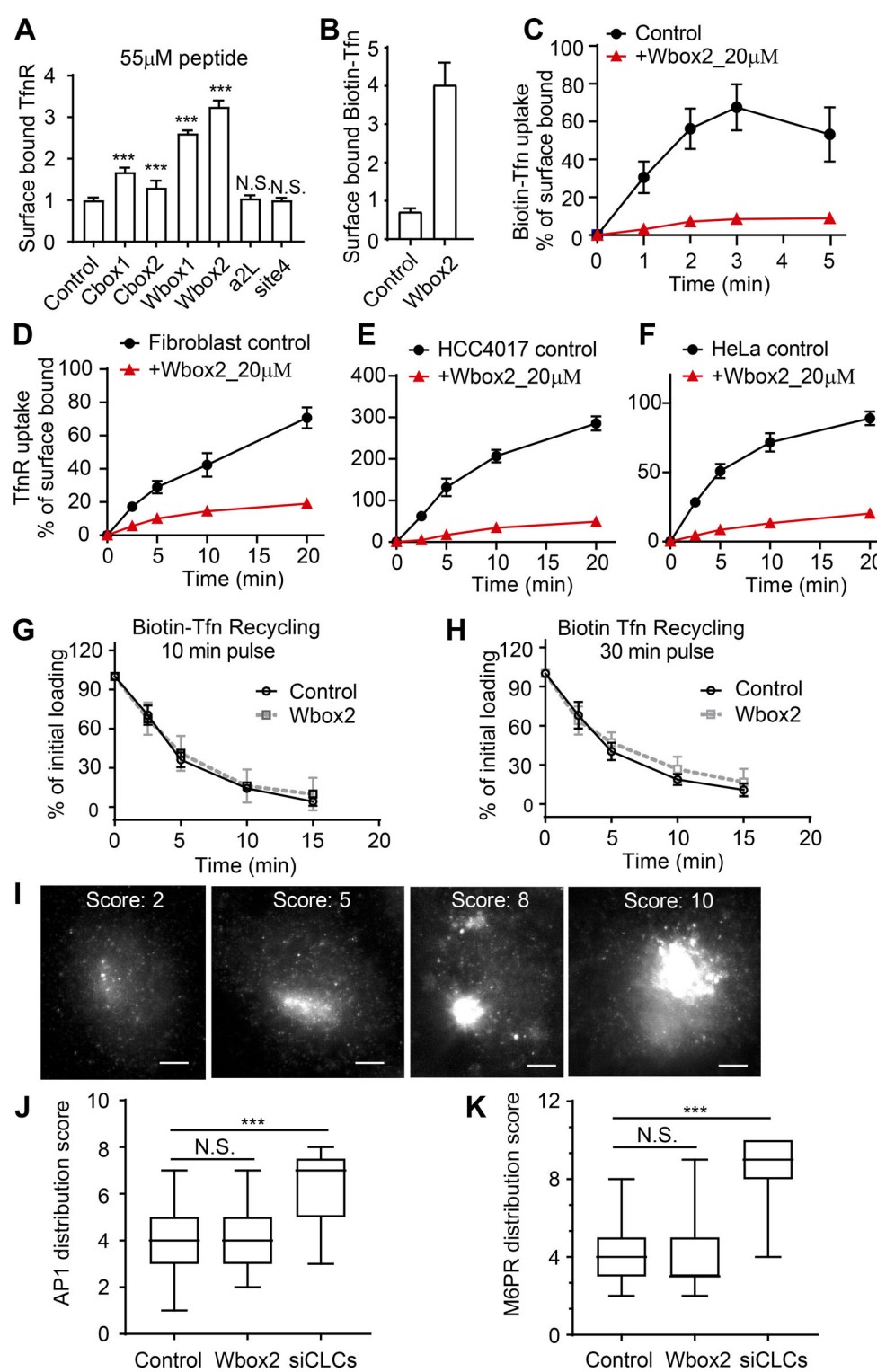

Figure S4. **Effect of TD-derived peptides on CME, Tfn recycling, and AP1-mediated Golgi trafficking. (A)** Effect of the inhibitory activities of the six peptides on the surface-bound TfnR. **(B and C)** Single-round Biotin-Tfn uptake revealed that Wbox2 strongly enhanced surface bound Biotin-Tfn (B) and reduced Biotin-Tfn uptake efficiency (C). **(D–F)** The Wbox2 peptide inhibited TfnR uptake in Fibroblast, HCC4017, and HeLa cells. **(G and H)** Effects of Wbox2 on Biotin-Tfn recycling after either 10- or 30-min pulse. **(I–K)** Immunostaining of AP1 γ-adaptin and M6PR in control HeLa cells as well as cells treated with Wbox2 (10 μM, preincubation for 30 min at 37°C) or siRNA knockdown CLCa+b (siCLCs). The AP1 and M6RP distribution in cells was quantified by grading them on the basis of degree of concentration at the perinuclear region: lowest score for completely dispersed phenotype, highest score for majority of signal being concentrated at perinuclear region, as representative images of M6PR shown in I. Scale bars = 10 μm. **(J and K)** Wbox2 treatment did not alter AP1 or M6PR distribution, whereas siCLCa+b as a positive control showed a strong effect by accumulating AP1 and M6PR in the perinuclear region. Number of cells quantified in J: 94 for control, 106 for Wbox2, and 33 for siCLCs. Number of cells quantified in K: 95 for control, 62 for Wbox2, and 37 for siCLCs. Error bars are SD. ***, P ≤ 0.001. N.S., not significant.

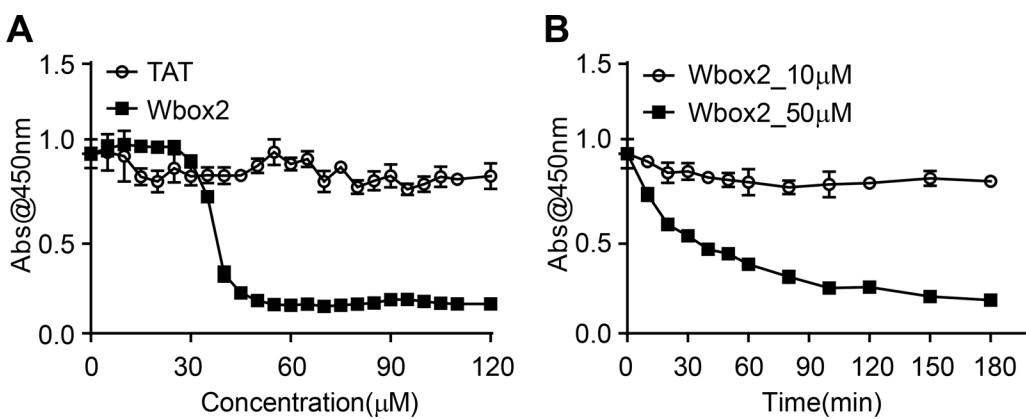

Figure S5. **Cell viability after Wbox2 treatment. (A)** Quantitation of viable cell number after treatment with varied concentrations of TAT and Wbox2 peptides using Cell Counting Kit 8 (CCK-8). **(B)** Quantification of viable cell number after treatment with 10 or 50 µM of Wbox2 and varied incubation times. Average data ± SD are from $n$ = 4 replicates.

Video 1. **Time-lapse TIRFM imaging of stable ARPE/HPV cells expressing eGFP-CLCa under control conditions.** Cells were infected with TDD and tTA-encoding adenoviruses but incubated in the presence of 100 ng/ml Tet to suppress TDD expression. Images were obtained at 1 frame/s and collected for 7.5 min. Video is accelerated 50-fold.

Video 2. **Time-lapse TIRFM imaging of stable ARPE/HPV cells expressing eGFP-CLC.** Cells were infected with TDD and tTA-encoding adenoviruses but incubated in the absence of Tet to induce TDD expression. Images were obtained at 1 frame/s and collected for 7.5 min. Video is accelerated 50-fold.

Video 3. **Time-lapse TIRFM imaging of stable ARPE cells expressing α-eGFP-AP2 under control conditions.** Cells were infected with TDD and tTA-encoding adenoviruses but incubated in the presence of 100 ng/ml Tet to suppress TDD expression. Images were obtained at 1 frame/s and collected for 7.5 min. Video is accelerated 50-fold.

Video 4. **Time-lapse TIRFM imaging of stable ARPE cells expressing α-eGFP-AP2.** Cells were infected with TDD and tTA-encoding adenoviruses but incubated in the absence of Tet to induce TDD expression. Images were obtained at 1 frame/s and collected for 7.5 min. Video is accelerated 50-fold.

Video 5. **Time-lapse TIRFM imaging of stable ARPE/HPV cells expressing eGFP-CLC 5 d after siRNA-mediated knockdown of SNX9.** Images were obtained at 1 frame/s and collected for 7.5 min. Video is accelerated 50-fold.

Video 6. **Time-lapse TIRFM imaging of stable in ARPE cells expressing α-eGFP-AP2 and mRuby-CLCa incubated in the presence of 20 µM Wbox2.** Left shows α-eGFP-AP2 channel and right is mRuby-CLCa channel. Images were obtained at 1 frame/s and collected for 7.5 min. Video is accelerated 50-fold.

**Provided online are three tables. Table S1 lists proteins involved in CME pathway from three independent trials. Table S2 lists proteins involved in membrane trafficking pathways from three independent trials. Table S3 lists reagents used in this work, including chemicals, proteins, antibodies, and siRNAs.**

