## [Peer Review File · The Journal of Cell Biology]

Wbox2: A clathrin terminal domain-derived peptide inhibitor of clathrin-mediated endocytosis

Zhiming Chen, Rosa Mino, Marcel Mettlen, Peter Michaely, Madhura Bhawe, Dana Reed, and Sandra Schmid

Corresponding Author(s): Sandra Schmid, UT Southwestern Medical Center

Review Timeline:

Submission Date:	2019-08-23
Editorial Decision:	2019-09-25
Revision Received:	2019-12-03
Editorial Decision:	2020-01-15
Revision Received:	2020-03-02
Editorial Decision:	2020-04-15
Revision Received:	2020-04-27

Monitoring Editor: Jodi Nunnari

Scientific Editor: Marie Anne O'Donnell

Transaction Report:

DOI: <https://doi.org/10.1083/jcb.201908189>

September 25, 2019

Re: JCB manuscript #201908189

Dr. Sandra L. Schmid
UT Southwestern Medical Center
6000 Harry Hines Blvd
Dallas, Texas 75390

Dear Dr. Schmid,

Thank you for submitting your manuscript entitled "CMEpi, a potent and selective structure-based inhibitor of clathrin-mediated endocytosis". The manuscript was assessed by expert reviewers, whose comments are appended to this letter. We invite you to submit a revision if you can address the reviewers' key concerns, as outlined here.

As you can see from the reviews below, all three reviewers are in favor of the idea that a specific inhibitor of clathrin-mediated endocytosis will be a very useful tool to cell biologists. However, there are still substantial concerns. Reviewer 1 disagrees with the conclusion that the new inhibitor CMEpi presented in this study is truly selective. All three reviewers found a lack of convincing mechanistic understanding of how the inhibitor works. In particular, reviewer 2 and 3 think it is important to at least demonstrate that the peptides are folded (also mentioned by reviewer 1) and bind to any endocytic proteins with a clathrin box such as AP2/SN9. We think the paper is potentially a better fit for the Tool format, which would normally require new cell biological insight obtained using the inhibitor to be demonstrated. However, considering the potential widespread application of these inhibitors and their use by non-specialist labs, we think it is more important in this case to ensure the highest possible standards when such easily accessible tools are introduced in the field, which we hope you would also agree. Given the interest towards a new specific inhibitor, I am happy to consider a substantially revised manuscript if the issues on specificity and mechanisms could be addressed.

GENERAL GUIDELINES:

Text limits: Character count for an Article is < 40,000, not including spaces. Count includes title page, abstract, introduction, results, discussion, acknowledgments, and figure legends. Count does not include materials and methods, references, tables, or supplemental legends.

Figures: Articles may have up to 10 main text figures. Figures must be prepared according to the policies outlined in our Instructions to Authors, under Data Presentation, <http://jcb.rupress.org/site/misc/ifora.xhtml>. All figures in accepted manuscripts will be screened prior to publication.

*****IMPORTANT:** It is JCB policy that if requested, original data images must be made available.

Failure to provide original images upon request will result in unavoidable delays in publication. Please ensure that you have access to all original microscopy and blot data images before submitting your revision.***

Supplemental information: There are strict limits on the allowable amount of supplemental data. Articles may have up to 5 supplemental figures. Up to 10 supplemental videos or flash animations are allowed. A summary of all supplemental material should appear at the end of the Materials and methods section.

The typical timeframe for revisions is three months; if submitted within this timeframe, novelty will not be reassessed at the final decision. Please note that papers are generally considered through only one revision cycle, so any revised manuscript will likely be either accepted or rejected.

Thank you for this interesting contribution to Journal of Cell Biology. You can contact us at the journal office with any questions, cellbio@rockefeller.edu or call (212) 327-8588.

Sincerely,

Min Wu, Ph.D.
Monitoring Editor

Marie Anne O'Donnell, Ph.D.
Scientific Editor

Journal of Cell Biology

Reviewer #1 (Comments to the Authors (Required)):

Chen et al. describe a new peptide based inhibitor of endocytosis (CMEpi). This peptide can enter cells and inhibit CME effectively and will prove a very useful tool to cell biologists, particularly as previous attempts at an inhibitor (e.g. pitstop) have failed. The paper is very interesting and overall good. The results are presented to convince the reader that CMEpi works in an analogous manner to overexpression of clathrin's terminal domain + distal region. In reality CMEpi could be working quite differently, and this part of the paper needs strengthening.

1. First of all, I am really surprised that a peptide which corresponds to some residues in TD acts in the same way as TD/TDD itself. The TD is a folded domain, whereas the peptide is likely unstructured. It is also surprising that the W-box and not the other binding sites can be mimicked to produce an inhibitor. Nevertheless, the peptide clearly inhibits endocytosis, but this raises the question of how (and whether it is by the same mechanism as TD/TDD). In several places the results are generalized: e.g. TD acts the same was as TDD, Wbox2 and Wbox1 are the same. However these points are not rigorously tested. For example, the section

entitled "Wbox2 is a potent and specific peptide inhibitor" relies on data obtained with only Wbox1 (Fig 7C+D). This should be repeated with Wbox2. On page 7, TDD results are extended to TD without experimental verification. On page 8 it says "TDD/TD [...] selectively inhibits CME" this is shown for TDD but not for TD. More cautious wording is appropriate especially if it turns out that CMEpi works in a different way than the authors think.

2. The authors claim CMEpi is "selective". On reading the paper, my conclusion is that TDD is selective but that CMEpi is not.

a) There is an effect on CIE with WBox2 (and not with TDD), so the authors cannot conclude in the title that the CMEpi is selective.

b) There is no functional test of Golgi traffic to show that CMEpi is selective for CME over other clathrin-mediated transport. Some images are shown (but not quantified) but what is needed is a functional assay such as cathepsin D export or RUSH.

c) Similarly the endosome distribution data is anecdotal.

d) Fig S6H There looks like there might be some effect on recycling of Tfn with WBox2 if the assay was allowed to continue.

3. A toxicity test with the peptide would be useful to future users of CMEpi. Especially given the high dosage required and the problems that pitstops had with toxicity which are mentioned in the introduction.

4. There is no control for the overexpression TDD. Maybe simply expressing any protein in this system inhibits CME. A great control here would be expression of TDD with the W-box residues mutated.

Reviewer #2 (Comments to the Authors (Required)):

This work shows the inhibitory effects on clathrin-mediated endocytosis of different short sequence peptides fused to a TAT motif then allowed to concentrate in the cell interior. The sequences correspond to different blades from the clathrin terminal domain β -propeller known to interact with specific effectors (AP2, SNX9, etc). Upon wash out, their effects are reversible.

Adding an inhibitory reagent 'specific' to clathrin endocytosis is a welcome step to the field. As such, I am in favor of publishing this work as a tool-box. Having said this, in my opinion it is essential that prior to publication the authors substantially tone down their statements and interpretations related to the structural aspects. This is important to ensure the highest possible standards for the field (and implicitly) to the authors.

If on the other hand, the author's wish to maintain their structural interpretations and conclusions, then they have to experimentally demonstrate that the synthetic peptide adopts the same structure (β -strands) as in a fully folded TD whether or not they had bound to the appropriate target binding motifs. Typically this would be an NMR study. Short of such demonstration, it is simply incorrect to state (as it is done all through out from the abstract to discussion and implicit in the title) that the membrane-penetrating peptides mimic the known binding sites on the TD of clathrin. Also essential is to experimentally demonstrate that mutant(s) that prevent folding of the β -strands (not involved in direct interaction with the target sequence peptide) also fail to interact.

Several times the authors state the effects of the membrane-permeable peptides might be explained by a direct interaction. The simplest explanation, however, is that membrane-bound peptide acts by

a squelching effect simply by capturing the target protein onto the membranes containing a very high concentration of the TD peptide, effectively by an avidity type of effect, particularly since the binding constants between TD and targets is roughly in the low - medium micromolar regime. This is apparent by increase of AP2 signal non-specifically associated with the pm, as described by the authors.

The authors also exploit their CME analysis to provide interpretation to the effects in ccp dynamics observed upon incubation of the cells with the membrane-penetrating peptides. I agree one should present quantitative analysis, at the same time, I feel the mechanistic interpretations to explain the inhibitory effects are over stated.

Besides these key points, the authors should properly make reference to work by others. I believe this is a matter of professional courtesy, and as we all know, lab members are very unhappy if proper attributions are missing.

Reviewer #3 (Comments to the Authors (Required)):

This paper is divided into two (tenuously connected) parts: The first part, classic detailed and comprehensive Schmid lab imaging work, dissects how overexpression of the CHC TDD protein in cultured cells impacts the CME machinery by disrupting normal AP2-CHC interactions at the PM. As this showed TDD disruption of clathrin recruitment to CCPs was most likely responsible for the inhibition, part two explored the use of novel TD based cell permeable inhibitors to bind to clathrin box sequences.

Despite over half a century of work on clathrin and CME, there is still no reliable, highly selective and universally applicable inhibitor of the process. If analysis of the TDD inhibition lead to the development of a trustworthy, easily used and highly selective inhibitor of CME, this is a big step forward.

However, I have some serious major concerns about the work presented:

In the second last line of the discussion they make a key claim: "Importantly, we detailed the mechanism of CMEpi inhibition, which illustrates the likely basis for its specificity." To me, the mechanistic description of how the TD derived peptides might bind to clathrin box peptides to interfere with CME is really opaque. It is well beyond the grasp of the casual or non specialist reader who would be most interested in using the peptides as tools to block CME. In fact, the bulk of the discussion explores the nature and mode of TDD inhibition to what is currently known about TD binding sites and CCP formation. This could have been reasonably predicted and, arguably, much more practically applicable is how the cell penetrating peptide work.

But something is critically missing from this work on the peptide inhibitors, that would link the two parts of the manuscript together and really strengthen the mechanistic claims about the inhibitor peptides. The authors need to show directly in biochemical assays that the TD derived peptide "mimics" they use as inhibitors in cells indeed bind physically to CME proteins with a clathrin box, like AP2 and SNX9. This missing data is VITAL to support mechanistic similarity to the TDD results in the first part of the paper. In addition, at the concentrations used on cells, do the peptide mimics interfere with AP2 and SNX9 binding to native clathrin trimers in direct biochemical assays? This type of direct biochemical analysis would reveal the molecular basis for the observed inhibition instead of relying on inferences.

Moreover, to some degree, the inhibitory peptide approach and the line of work is anathema to traditional structure-function analysis and the widely held precept that structure underlies function. The 4 stranded TD blades are to some degree autonomous, but the different blades come together in space to form structured 3D contact surfaces for clathrin box sequences. They report that very short isolated bits of the TD propellers appear to be selective competitive inhibitors. If this works, why does the TD have to be folded into a 7 bladed propeller at all? As I understand it, even non contiguous sets of TD amino acids that were previously pinpointed to be important by structural studies, when presented as part of a 10 aa peptide fused to TAT, can act as CME inhibitors. No background on the normal binding of clathrin boxes by the TD is presented. Jim Keen's group showed a long time ago that the first 100 residues of the TD, which fold up into blades 1 & 2 of the propeller, can bind to clathrin box sequences. This is not mentioned, nor is why 10 amino acids are as good as 100.

Worrying is the inhibition of CD44 and CD59 uptake by Wbox1/2 peptide application (Fig 7). The authors state in the results that a previous microscopy based study suggested these proteins might undergo CME (middle of page 9), but there is no citation to that work. More perplexing is that it seems their own data in Fig 2 contradicts this point of view. In the same cells, they show no inhibition by the TDD?

Tim Ryan's group showed previously that a steep threshold for clathrin required for CME (Moskowitz, et al MBOC 2005). This clearly should be discussed. Along these lines, this work should be related to the "hot wiring" of CCPs previously published in the JCB. . Overexpression of the "hooks" should deplete clathrin from normal sites of CME so hot wiring seems to represent sequestration of some of the cellular clathrin pool at irrelevant sites. But Royle and his people report that this does not interfere with the function of clathrin and inhibit Tf uptake. It is essential to clarify this.

On page 11, near the end of the first paragraph, the statement: "The stronger affinity of AP2 for clathrin may reflect the existence of a second binding site on the b1-appendage domain and the bipartite nature of AP2-clathrin interactions." What does this mean exactly?

Also on page 11, the third paragraph tries to rationalize why the GST-TD binding assays did not identify EAPs, and the authors wonder whether EAPs only bind clathrin with low affinity. But it is well excepted and reproduced by many groups that in the reverse experiment, using GST-EAPs, clathrin box regions from EAPs bind assembled clathrin triskelia from soluble cytosol extracts with good affinity. This underscores the point that it is the inherent trivalency of the triskelion that is necessary for the stable association with clathrin, given the low micromolar affinity of clathrin box peptides for the TD. The basic reason they don't see good EAP binding to the over expressed TD in IPs is it is monomeric.

Minor comments:

The authors report that the TDD associates with the PM. What makes TDD bind to the PM if it is NOT localized to CCPs?

On the basis of IPs one of the conclusions regarding how TDD overexpression works is that it binds to (soluble) AP2 and SNX9. How does TDD bind to soluble AP2 if it is in a locked conformation with the beta2 adaptin clathrin box sequence not accessible?

Sandra L. Schmid, Ph.D.
Cecil H. Green Distinguished Chair
in Cellular and Molecular Biology
Professor & Chairman

Department of Cell Biology

December 3, 2019

Min Wu, Ph.D.
Monitoring Editor
JCB

Dear Dr. Wu:

We are grateful to you and the reviewers for your efforts and comments that have helped to strengthen our manuscript. We have addressed the reviewers' concerns by conducting further experiments and clarifying statements in the text, especially those that might have 'overstated' our findings. Most significantly, we have conducted biophysical (isothermal calorimetry) and biochemical (pulldown assays) to show that CMEpi, the most potent TD peptide-based inhibitor indeed interacts directly and with high affinity to AP2 and SNX9 (new Figure 9). Reviewer 1 stated that '*I am really surprised that a peptide which corresponds to some residues in TD acts in the same way as TD/TDD itself*': as were we. Thus, one of the most important take-homes of our paper, although well beyond its scope, is that the current dogma regarding clathrin-AP and clathrin-EAP interactions might need to be revisited.

Below we address the referees' comments (in *italic*) point by point. Key changes in the text and figure legends have been highlighted in yellow. We hope that you and the referees will now find this revised manuscript suitable for publication in JCB.

Reviewer #1:

Chen et al. describe a new peptide-based inhibitor of endocytosis (CMEpi). This peptide can enter cells and inhibit CME effectively and will prove a very useful tool to cell biologists, particularly as previous attempts at an inhibitor (e.g. pitstop) have failed. The paper is very interesting and overall good. The results are presented to convince the reader that CMEpi works in an analogous manner to overexpression of clathrin's terminal domain + distal region. In reality CMEpi could be working quite differently, and this part of the paper needs strengthening.

1. First of all, I am really surprised that a peptide which corresponds to some residues in TD acts in the same way as TD/TDD itself. The TD is a folded domain, whereas the peptide is likely unstructured. It is also surprising that the W-box and not the other binding sites can be mimicked to produce an inhibitor. Nevertheless, the peptide clearly inhibits endocytosis, but this raises the question of how (and whether it is by the same mechanism as TD/TDD).

Response: As stated above we have added new, quantitative data (Figure 9) showing that CMEpi binds directly to SNX9 and inhibits TD-AP2 interactions with concentrations comparable to its IC₅₀ for CME. These data support our conclusion that CMEpi inhibits CME by a similar mechanism as TDD/TD.

In several places the results are generalized: e.g. TD acts the same as TDD, Wbox2 and Wbox1 are the same. However these points are not rigorously tested. For example, the section entitled "Wbox2 is a potent and specific peptide inhibitor" relies on data obtained with only Wbox1 (Fig 7C+D). This should be repeated with Wbox2. On page 7, TDD results are extended to TD without experimental verification. On page 8 it says "TDD/TD [...] selectively inhibits CME" this is shown for TDD but not for TD. More cautious wording is appropriate especially if it turns out that CMEpi works in a different way than the authors think.

Response: We have repeated experiments on Figure 7C+D with Wbox2. The results showed that Wbox2 is also fast-acting in CME inhibition and the inhibition effect is reversible. We have also been careful to avoid generalizations based on data not presented or experiments not performed. We have now focused all of our detailed characterization and conclusions on the effects of WBox2, aka CMEpi.

Changes:

1. Figure 7C+D, Wbox1 data have been replaced with Wbox2 data.
2. 'TDD/TD' has been changed to 'TDD' throughout the manuscript.

2. The authors claim CMEpi is "selective". On reading the paper, my conclusion is that TDD is selective but that CMEpi is not.

a) There is an effect on CIE with WBox2 (and not with TDD). so the authors cannot conclude in the title that the CMEpi is selective.

Response and Changes: We have now addressed this concern in a paragraph in the expanded discussion. Our interpretation is not that CMEpi is not selective, but the putative CIE markers, the GPI-anchored protein CD59 and the transmembrane protein CD44 can also be CME cargo. We cite previous studies by us (Damke et al., 1995) and others (Bitsikas et al., 2014) showing that CIE is up-regulated after prolonged inhibition of CME. We believe the difference between the effects of CMEpi and TDD overexpression relate to acute v. prolonged effects. CMEpi will be a valuable tool to dissect mechanisms of endocytosis. Note that dynamin is required for both and hence DN-Dyn or Dynasore inhibits both CME and some CIE pathways.

b) There is no functional test of Golgi traffic to show that CMEpi is selective for CME over other clathrin-mediated transport. Some images are shown (but not quantified) but what is needed is a functional assay such as cathepsin D export or RUSH.

Response: We have conducted immunofluorescence of AP1 and mannose-6-phosphate-receptor (M6PR) in control cells as well as cells treated with CMEpi or siRNA knockdown clathrin light chains (a+b), as a positive control. We further quantified the AP1 and M6PR

distribution in cells by grading them on the basis of degree of concentration at the perinuclear region. Lowest score for completely dispersed phenotype, highest score for majority of signal being concentrated at perinuclear region. The results indicate that CMEpi treatment did not alter AP1 or M6PR distribution, while siCLCa+b showed a strong effect by accumulating AP1 and M6PR at perinuclear region. From this, we concluded that CMEpi treatment does not affect AP1-mediated cargo trafficking from the Golgi.

Changes: In Figure S4, we have added the quantified AP1 and M6RP distribution data as (I-K).

c) Similarly the endosome distribution data is anecdotal.

Response and changes: We agree that the endosome distribution data is not helpful in telling our story, thus we have removed this data from our figure and text.

d) Fig S6H There looks like there might be some effect on recycling of Tfn with WBox2 if the assay was allowed to continue.

Response: Thank you for pointing this out. We have repeated Tfn recycling with Wbox2 twice and confirmed that Tfn recycling is not significantly impaired by CMEpi treatment. By removing biotin from the culture media we increase our signal:noise on these assays that measure recycling of biotinylated-Tfn providing cleaner and more reproducible results. We noted this methodological change in the methods.

Changes: Figure S6G+H have been replaced with updated Tfn recycling data, and now as Figure S4G+H.

3. A toxicity test with the peptide would be useful to future users of CMEpi. Especially given the high dosage required and the problems that pitstops had with toxicity which are mentioned in the introduction.

Response and changes: We thank the reviewer for this suggestion. We have now conducted toxicity tests at varied concentrations of CMEpi and varied incubation times (Figure S5). There is an ~10-fold difference in concentrations of CMEpi that inhibit CME vs those that show cytotoxicity.

4. There is no control for the overexpression TDD. Maybe simply expressing any protein in this system inhibits CME. A great control here would be expression of TDD with the W-box residues mutated.

Response and changes Numerous studies by the Schmid lab (and others) have shown that overexpressing many proteins, e.g.. WT dynamin, CLCs (Aguet et al., 2013) does not inhibit CME. Given the in vivo data from Lemmon and Royle's labs, we do not anticipate that disrupting any of the individual sites will ablate the inhibition. However, we now show (Fig. 7F) that mutating the critical residues in CMEpi significantly reduces its ability to inhibit CME.

Reviewer #2:

This work shows the inhibitory effects on clathrin-mediated endocytosis of different short sequence peptides fused to a TAT motif then allowed to concentrate in the cell interior. The sequences correspond to different blades from the clathrin terminal domain β -propeller known to interact with specific effectors (AP2, SNX9, etc). Upon wash out, their effects are reversible.

Adding an inhibitory reagent 'specific' to clathrin endocytosis is a welcome step to the field. As such, I am in favor of publishing this work as a tool-box. Having said this, in my opinion it is essential that prior to publication the authors substantially tone down their statements and interpretations related to the structural aspects. This is important to ensure the highest possible standards for the field (and implicitly) to the authors.

If on the other hand, the author's wish to maintain their structural interpretations and conclusions, then they have to experimentally demonstrate that the synthetic peptide adopts the same structure (β -strands) as in a fully folded TD whether or not they had bound to the appropriate target binding motifs. Typically, this would be an NMR study. Short of such demonstration, it is simply incorrect to state (as it is done all through out from the abstract to discussion and implicit in the title) that the membrane-penetrating peptides mimic the known binding sites on the TD of clathrin. Also essential is to experimentally demonstrate that mutant(s) that prevent folding of the β -strands (not involved in direct interaction with the target sequence peptide) also fail to interact.

Response and changes: These comments essentially parallel the 'surprise' expressed by referee 1. We do not believe that a 10 aa peptide adopts the same conformation as the β -propeller surface. However, we now provide strong evidence (described below) that CMEpi indeed functions by interfering with TD-AP2 and TD-SNX9 interactions, presumably by binding to and masking the small linear motifs (SLiMs) on AP2 and SNX9 that interact with TD. We have also elaborated in the discussion on the rational of CMEpi inhibition as described below. We hope the editor and referee agree that NMR structural studies are beyond the scope of this manuscript.

The amino acids encompassing the sequence of CMEpi are colored in orange above. These residues demarking the W-box site are located across several blades of the TD β -propeller.

Thus the β -propeller fold serves to generate a top-central interaction surface, which is presumably mimicked by our CMEpi design. Thus, the synthesized CMEpi is unlikely to form a β -strand structure when interacting with EAPs.

We have designed a mutant peptide in which all the reported key residues were replaced by alanines (F27/Q152/I154/I170A). Compared to CMEpi, this mutant showed greatly reduced inhibition of CME. This data is now incorporated into Figure 7F.

Moreover, we have used ITC to directly measure the interactions of CMEpi peptide with purified SNX9 and AP2- β -hinge+appendage domain. The results indicate that CMEpi can indeed bind to SNX9 and AP2- β -hinge+appendage domain. Our further GST-TD pulldown assay using mouse brain extracts also showed that the addition of CMEpi inhibits the interaction between AP2 and TD. These results are now presented as Figure 9 and strengthen our statement that CMEpi peptide mimics the W-box site. They also suggest that the nature of interactions of both AP2 and SNX9 with TD, which are based on crystal structures with small peptides, might need to be revisited using larger portions of both proteins. Of course, this too is beyond the scope of this paper.

Several times the authors state the effects of the membrane-permeable peptides might be explain by a direct interaction. The simplest explanation, however, is that membrane-bound peptide acts by a squelching effect simply by capturing the target protein onto the membranes containing a very high concentration of the TD peptide, effectively by an avidity type of effect, particularly since the binding constants between TD and targets is roughly in the low - medium micromolar regime. This is apparent by increase of AP2 signal non-specifically associated with the pm, as described by the authors.

Response: Thanks to the reviewer for bringing up this possibility. In addition to the new results describe above, to answer this question, we have synthesized and applied Cy5 labeled CMEpi peptides (20 μ M for 30min) to ARPE-HPV cells with stable eGFP-CLCa expression. Under fluorescence microscope, CMEpi was not observed to accumulate on the plasma membrane, but in the interior of cells (see images shown below).

The authors also exploit their CME analysis to provide interpretation to the effects in ccp dynamics observed upon incubation of the cells with the membrane-penetrating peptides. I

agree one should present quantitative analysis, at the same time, I feel the mechanistic interpretations to explain the inhibitory effects are over stated.

Response and changes: To strengthen our mechanistic interpretations, we have conducted biochemical measurements (ITC and pulldown assay) to show that CMEpi interferes with the interactions between AP2-clathrin and SNX9-clathrin.

Besides these key points, the authors should properly make reference to work by others. I believe this is a matter of professional courtesy, and as we all know, lab members are very unhappy if proper attributions are missing.

Response: Thank you for pointing this out as we could not agree more with the importance of properly referencing others' work. We endeavored to more thoroughly reference papers on which our work is based or those that provide precedence for our findings. However, without specific examples, we may still have missed some and would be pleased to add them.

Reviewer #3:

This paper is divided into two (tenuously connected) parts: The first part, classic detailed and comprehensive Schmid lab imaging work, dissects how overexpression of the CHC TDD protein in cultured cells impacts the CME machinery by disrupting normal AP2-CHC interactions at the PM. As this showed TDD disruption of clathrin recruitment to CCPs was most likely responsible for the inhibition, part two explored the use of novel TD based cell permeable inhibitors to bind to clathrin box sequences.

Despite over half a century of work on clathrin and CME, there is still no reliable, highly selective and universally applicable inhibitor of the process. If analysis of the TDD inhibition lead to the development of a trustworthy, easily used and highly selective inhibitor of CME, this is a big step forward.

However, I have some serious major concerns about the work presented:

Response: Thanks for these comments and your careful reading of the manuscript. We agree that a trustworthy and selective inhibitor of CME would be a valuable contribution to the field. We hope that our further experiments encouraged by the referees have addressed your 'major concerns'

In the second last line of the discussion they make a key claim: "Importantly, we detailed the mechanism of CMEpi inhibition, which illustrates the likely basis for its specificity." To me, the mechanistic description of how the TD derived peptides might bind to clathrin box peptides to interfere with CME is really opaque. It is well beyond the grasp of the casual or non-specialist reader who would be most interested in using the peptides as tools to block CME. In fact, the bulk of the discussion explores the nature and mode of TDD inhibition to what is currently known about TD binding sites and CCP formation. This could have been reasonably predicted and, arguably, much more practically applicable is how the cell penetrating peptide work

Response and changes: In addition to further experiments described below, we have significantly revised the Discussion to include our interpretation of how CMEpi inhibits CME.

But something is critically missing from this work on the peptide inhibitors, that would link the two parts of the manuscript together and really strengthen the mechanistic claims about the inhibitor peptides. The authors need to show directly in biochemical assays that the TD derived peptide "mimics" they use as inhibitors in cells indeed bind physically to CME proteins with a clathrin box, like AP2 and SNX9. This missing data is VITAL to support mechanistic similarity to the TDD results in the first part of the paper. In addition, at the concentrations used on cells, do the peptide mimics interfere with AP2 and SNX9 binding to native clathrin trimers in direct biochemical assays? This type of direct biochemical analysis would reveal the molecular basis for the observed inhibition instead of relying on inferences.

Response and changes: Thank you for this helpful suggestion. We have measured the direct interactions of CMEpi peptide with AP2 and SNX9 using ITC. CMEpi was observed to interact with SNX9 (with potential two binding sites) and the AP2- β 2-hinge+appendage domain with micromolar-scale affinity. We have also conducted a GST-TD pulldown assay using mouse brain extract (which contains enriched AP2), and observed that addition of CMEpi inhibited the AP2-TD interactions. These results are now presented in Figure 9 as evidence for the mechanistic interpretation of CMEpi.

Moreover, to some degree, the inhibitory peptide approach and the line of work is anathema to traditional structure-function analysis and the widely held precept that structure underlies function. The 4 stranded TD blades are to some degree autonomous, but the different blades come together in space to form structured 3D contact surfaces for clathrin box sequences. They report that very short isolated bits of the TD propellers appear to be selective competitive inhibitors. If this works, why does the TD have to be folded into a 7 bladed propeller at all? As I understand it, even non-contiguous sets of TD amino acids that were previously pinpointed to be important by structural studies, when presented as part of a 10 aa peptide fused to TAT, can act as CME inhibitors. No background on the normal binding of clathrin boxes by the TD is presented. Jim Keen's group showed a long time ago that the first 100 residues of the TD, which fold up into blades 1 & 2 of the propeller, can bind to clathrin box sequences. This is not mentioned, nor is why 10 amino acids are as good as 100.

Response: We appreciated the reviewer's comment that motivate us to think about how the 10aa W-box site mimetic peptide can inhibit CME. The reviewer comments raised an interesting question: "why does the TD have to be folded into a 7 bladed propeller?" The W-box site is on the top center of TD domain, a composite structure/effect of different blades, and it is beyond the beta-strand structure. The result/purpose of bringing 7 blades together might be to generate a hot spot for interactions and regulation. Also, as stated above in response to referee 2, our results beg further structural studies on the nature of interactions between SliMs, which are typically present in multiple copies within long disordered domains and the TD β -propeller. Our current assumptions are based on small peptide-TD structures.

Worrying is the inhibition of CD44 and CD59 uptake by Wbox1/2 peptide application (Fig 7). The authors state in the results that a previous microscopy based study suggested these proteins might undergo CME (middle of page 9), but there is no citation to that work. More perplexing is that it seems their own data in Fig 2 contradicts this point of view. In the same cells, they show no inhibition by the TDD?

Response and change: Referee 1 raised this same issue. We have now addressed this concern in a paragraph in the expanded discussion. Our interpretation is not that CMEpi is no selective, but the putative CIE markers, the GPI-anchored protein CD59 and the transmembrane protein CD44 can also be CME cargo. We cite previous studies by us (Damke et al., 1995) and others (Bitsikas et al., 2014) showing that CIE is up-regulated after prolonged inhibition of CME. We believe the difference between the effects of CMEpi and TDD overexpression relates to acute v. prolonged effects. CMEpi will be an invaluable tool to dissect mechanisms of endocytosis. Note that dynamin is required for both and hence DN-Dyn or Dynasore inhibits both CME and some CIE pathways.

Tim Ryan's group showed previously that a steep threshold for clathrin required for CME (Moskowitz, et al MBOC 2005). This clearly should be discussed. Along these lines, this work should be related to the "hot wiring" of CCPs previously published in the JCB. Overexpression of the "hooks" should deplete clathrin from normal sites of CME so hot wiring seems to represent sequestration of some of the cellular clathrin pool at irrelevant sites. But Royle and his people report that this does not interfere with the function of clathrin and inhibit Tf uptake. It is essential to clarify this.

Response and change: We have now cited both Ryan's and Royle's interesting papers. The Royle paper demonstrated the sufficiency of the β 2-hinge ear to trigger clathrin recruitment. We were also surprised that CME could be triggered by simply recruiting clathrin to the PM in an AP2-independent manner, but the results presented were convincing. It is not clear how our current study can clarify the referee's concerns regarding Royle's work.

On page 11, near the end of the first paragraph, the statement: "The stronger affinity of AP2 for clathrin may reflect the existence of a second binding site on the b1-appendage domain and the bipartite nature of AP2-clathrin interactions." What does this mean exactly?

Response and change: We have clarified this in a new paragraph in the Discussion regarding the nature of clathrin-AP2 interactions and cited relevant papers.

Also on page 11, the third paragraph tries to rationalize why the GST-TD binding assays did not identify EAPs, and the authors wonder whether EAPs only bind clathrin with low affinity. But it is well excepted and reproduced by many groups that in the reverse experiment, using GST-EAPs, clathrin box regions from EAPs bind assembled clathrin tirskeia from soluble cytosol extracts with good affinity. This underscores the point that it is the inherent trivalency of the triskelion that is necessary for the stable association with clathrin, given the low micromolar affinity of clathrin box peptides for the TD. The basic reason they don't see good

EAP binding to the over expressed TD in IPs is it is monomeric.

Response and change: We agree and have now more clearly stated this in the Discussion.

Minor comments:

The authors report that the TDD associates with the PM. What makes TDD bind to the PM if it is NOT localized to CCPs?

Response: TDD binds to the PM mainly due to interaction with membrane-bound, active AP2. More AP2 on PM was observed upon TDD overexpression (Figure 5C-E).

On the basis of IPs one of the conclusions regarding how TDD overexpression works is that it binds to (soluble) AP2 and SNX9. How does TDD bind to soluble AP2 if it is in a locked conformation with the β 2 adaptin clathrin box sequence not accessible?

Response: We are well aware of David Owen's elegant structural studies on AP2 conformations and allosteric activation. These interactions are no-doubt dynamic and at high concentrations free TD can compete to open the AP2 complex. The consequences of this are both stabilization of AP2 complexes on the PM independent of full length clathrin (note that 'stabilization' is a relative term as these clathrin-deficient AP2 puncta a very short-lived) and interactions with the cytosolic pool. These distinct pools of AP2-TD complexes are no doubt in dynamic equilibrium.

References:

Aguet, F., C.N. Antonescu, M. Mettlen, S.L. Schmid, and G. Danuser. 2013. Advances in analysis of low signal-to-noise images link dynamin and AP2 to the functions of an endocytic checkpoint. *Dev Cell*. 26:279-291.

Sincerely yours,

Sandra L. Schmid, Ph.D.
Professor and Chairman

January 13, 2020

Re: JCB manuscript #201908189R

Dr. Sandra L. Schmid
UT Southwestern Medical Center
6000 Harry Hines Blvd
Dallas, Texas 75390

Dear Dr. Schmid,

Thank you for submitting your revised manuscript entitled "CMEpi, a potent and selective structure-based inhibitor of clathrin-mediated endocytosis" to the Journal of Cell Biology. The manuscript has been assessed by the original reviewers, whose reports are appended below (Rev#2 was not available).

As you can see from the comments of Reviewer #3, and I concur, these experiments were somewhat preliminary to conclude some of the mechanistic claims. Second, I am not convinced that CMEpi is a "selective" clathrin-mediated endocytosis inhibitor. Given its clear inhibitory effect on CD44 and CD59 uptake, CMEpi would likely suffer from the same criticism that pitstops received for non-specificity. The explanation provided in the discussion "CIE is up-regulated after prolonged inhibition of CME" contradicts with the actual observation (inhibition of CIE, instead of up-regulation). In addition, if CD44 and CD59 are in fact partially CME cargoes, the earlier conclusion that TDD is specific becomes questionable. As such, it is not clear whether CMEpi as a tool to dissect mechanisms of endocytosis will be definitive on its own, which is what one expects from a specific inhibitor.

I continue to think this paper provides an interesting body of work that explores the interaction of clathrin TD with a different motif from the existing inhibitors. However, with these concerns, I am afraid that the two major conclusions of the manuscript: "a membrane-penetrating peptide, CMEpi, that potently and selectively inhibits CME" and "CMEpi provides a potent new tool to inhibit CME via a molecularly defined mechanism" are not supported by its data.

JCB policy is that papers are considered through only one major revision cycle. Given the significant remaining reviewer concerns, we unfortunately cannot offer publication of the manuscript.

If you would like to transfer your reviewer comments to another journal for consideration elsewhere, please contact the journal office and we would be happy to arrange the transfer on your behalf, cellbio@rockefeller.edu or call (212) 327-8588.

We appreciate the effort that has gone into the revisions and regret that the outcome is not more positive.

Sincerely,

Min Wu, Ph.D.
Monitoring Editor

Marie Anne O'Donnell, Ph.D.
Scientific Editor

Journal of Cell Biology

Reviewer #1 (Comments to the Authors (Required)):

The authors have addressed the comments that we raised satisfactorily (with the exception of the below).

I raised the issue below confidentially with the Editor during the first round of review. However, the affected figures remain unchanged in the revised version. I think the journal and authors should address this issue before acceptance.

"We noticed two irregularities in the western blots in this paper and would appreciate it if the journal could use their image integrity checks on them. When we adjusted the contrast on the blots in Fig. 2A and Fig S3A, we noticed that in FigS3A it appears that a band may have been removed and in Fig 2A the blot is presented in a misleading way so as to obscure the leaky expression of TDD (bands can clearly be seen for 5-20 ng/ml after adjusting the contrast). We have not mentioned this in our report."

Reviewer #3 (Comments to the Authors (Required)):

Again, this study is composed of two parallel arms. The first is solid, but rather pedestrian and well studied previously (this is cited in the text). It is the second arm (the identification of CHC TD derived cell permeant inhibitors of CME) that would justify acceptance at JCB in my view.

The authors have now responded to the critical issue of whether the isolated peptide pieces of the TD propeller interact directly with AP2 and/or EAPs. Given the measured Kds reported in the all new Figure 9, it appears that the principal concern of all three reviewers has been clearly addressed experimentally.

However, i am unfortunately (still) not convinced.

First there are no negative controls for the ITC experiments, nor peptide alone traces. From the legend, it is impossible to figure out exactly what has been done in panel A. Comparative traces with the mutated (inactivated) CMEpi are necessary.

The same is true for the data presented in panel B of Figure 9. In this pulldown with a highly enriched source of AP2 there is no negative control. Also, there is missing the effect of mutating the key residues on the inhibition by the CMEpi peptide. This experiment does not show direct binding of EAPs to ether the bound Gst-TD, AP2 or the soluble (but totally unquantified in this work) EAPs in the extract..

What is the conclusion from the right part of panel A and the results on panel B that supposedly explains the incomplete binding of the CMEpi peptide seen in the ITC runs?

Overall, Panel B presents a complicated experiment and the side by side use of a strong TD-ear

and a TD--EAP inhibitor should be considered for comparison.

The absence of these essential controls makes the results open to question. In this reviewer's mind, the matter is critical given the mechanistic claims and novelty prominence of the inhibitory very short TD peptides in this manuscript.

Second, because the K_d s are in the range of typical EAP interactions, an attempt to decipher whether the inhibitory sequences from the TD have any meaningful similarity to known EAP binding sequences (eg. role/positioning of hydrophobic residues) should be undertaken.

JCB20198189R

Response to editors concerns:

The manuscript has been assessed by the original reviewers, whose reports are appended below (Rev#2 was not available).

It is unfortunate that rev#2 was unavailable, as I believe we were able to address his/her concerns. Indeed, in their original comments reviewer 2 stated that *"Adding an inhibitory reagent 'specific' to clathrin endocytosis is a welcome step to the field. As such, I am in favor of publishing this work as a tool-box. Having said this, in my opinion it is essential that prior to publication the authors substantially tone down their statements and interpretations related to the structural aspects.* In response, we had better explained the design of our inhibitory peptides and why we did not expect our peptides to adopt a TD-like β -strand structure as the reviewer correctly pointed out. The new ITC data and our biochemical demonstration that CMEpi indeed inhibits AP2-TD interactions further strengthens our interpretation as to the peptide's mechanism of action. Perhaps this reviewer might be persuaded to weigh in on this rebuttal, especially as reviewer 1 indicated that his/her concerns regarding the specificity and mechanism of inhibition were satisfactorily addressed.

As you can see from the comments of Reviewer #3, and I concur, these experiments were somewhat preliminary to conclude some of the mechanistic claims.

See comments below. While we may not have all the answers, we clearly demonstrate, by the most rigorous and direct method—ITC—that CMEpi directly binds to both AP2 and SNX9 with Kds in the range of its inhibitory effects on CME. We also show biochemically that CMEpi inhibits TD-AP2 interactions, again at concentrations consistent with its effects on CME. Together these biochemical properties fully explain the effects we see on CME and CCP dynamics. Thus, we do not believe our mechanistic claims are unsupported by the data.

Second, I am not convinced that CMEpi is a "selective" clathrin-mediated endocytosis inhibitor. Given its clear inhibitory effect on CD44 and CD59 uptake, CMEpi would likely suffer from the same criticism that pitstops received for non-specificity.

Importantly, our new finding that CMEpi directly binds SNX9 with high affinity at two sites could also explain the effects of CMEpi on a subset of other clathrin-independent pathways, which we failed to consider. Indeed, we have shown in the past that siRNA knockdown of SNX9 inhibits CD44 uptake (Bendris et al., MBoC 2016) and that SNX9 colocalizes with GPI-anchored proteins in endocytic tubular structures (Yarar et al., 2007). We have now described these results in our Discussion and refer to our inhibitor, simply as Wbox2 to more accurately reflect its design. I would still argue that our peptide inhibitor is a more robust and specific reagent to inhibit different modes of endocytosis than either dynamin-inhibitors or Pitstop, as both have been shown to have off-target effects completely independent of their effects on any endocytic pathway. We have now revised our discussion to more directly compare the utility of Wbox2 relative to other small molecule, acute inhibitors of endocytosis for the benefit of future users. While, Wbox2 is not perfect, we believe it is a significant improvement over currently available acute inhibitors of endocytosis. Moreover, our demonstration of the potency of TAT-based peptides as inhibitors of CME inspires us, and hopefully others, to develop even more specific peptide-based inhibitors in the future.

The explanation provided in the discussion "CIE is up-regulated after prolonged inhibition of CME" contradicts with the actual observation (inhibition of CIE, instead of up-regulation). In addition, if CD44 and CD59 are in fact partially CME cargoes, the earlier conclusion that TDD is specific becomes questionable.

We were obviously still unclear in our explanation as your interpretation is incorrect. Indeed, it still remains a possibility that CIE pathways are upregulated after prolonged inhibition, either by overexpression of dominant-negative constructs or following prolonged siRNA knockdown, but not upregulated after acute inhibition, as with our peptides (or Pitstop). Our observation that neither CD44 nor CD59 uptake is inhibited after prolonged inhibition with TDD, but both are inhibited after acute inhibition with CMEpi is consistent with induction of compensatory pathways that occurs only after prolonged inhibition (as we and others have previously shown). Such a compensatory upregulation of CIE would restore the endocytic efficiency of CD44 and CD59 in long-term TDD-treated cells, but would not be able to restore CD44 and CD59 uptake after acute inhibition, as we have observed. We have edited this discussion for clarity.

That said, because neither TDD nor CMEpi inhibits fluid phase uptake of HRP, either the CD44 and CD59 pathways do not contribute significantly to bulk endocytosis (as had been suggested by Parton, Mayor and colleagues for the CLIC/GEEC pathway) or CD44 and CD59 are partially internalized by clathrin-mediated endocytosis in these cells. Confounding this discussion is the fact that the magnitude of these mechanistically distinct CIE pathways differs in different cell lines, as we have shown by directly measuring them in several nonsmall cell lung cancer cells lines (Elkin et al., 2015). It is also clear from Kirsten Sandvig's work that 'CIE' cargo like shiga toxin can be primarily internalized by CME in some cells (e.g. PMID 16098193), but by CIE pathways in others (Romer et al. PMID 18046403). Another example worth discussing is that while Parton, Mayor and colleagues have shown that CD44 uptake is not inhibited by prolonged DN-dynamin expression, others have published that it is inhibited by acute treatment with dynasore (Takahashi et al., PMID 25783601). Whether this represents another example of the differences between acute inhibition and prolonged inhibition leading to upregulation of compensatory CIE, or an off-target effect of dynasore is yet unclear. Finally, Nichols and colleagues showed that >90% of GPI-anchored cargos are internalized along with Tfn in nascent CCPs in HeLa, Cos and RPE cells (Bitsikas et al., PMID 25232658). The discussion of these uncertainties, now included in our revised manuscript, would be beneficial to the field, especially as the mechanisms underlying CIE remain enigmatic.

As such, it is not clear whether CMEpi as tool to dissect mechanisms of endocytosis will be definitive on its own, which is what one expect from a specific inhibitor.

Thanks for your prodding, you are correct. On its own CMEpi may not be a definitive tool to dissect all mechanisms of endocytosis, but it is able to distinguish AP2 and SNX9-dependent mechanisms from bulk fluid-phase uptake. Given the uncertain specificity, yet broad use of dynamin inhibitors, and the nonspecific cytotoxicity and off-target effects of Pitstops we strongly believe that our peptide-based inhibitor remains a valuable addition to the endocytosis tool chest.

I continue to think this paper provides an interesting body of work that explores the interaction of clathrin TD with a different motif from the existing inhibitors.

Thank you. In addition to introducing a new class of endocytosis inhibitors, we also provide the first direct evidence that TD-EAP interactions function through late stages of CME among other novel contributions, as listed above.

However, with these concerns, I am afraid that the two major conclusions of the manuscript: "a membrane-penetrating peptide, CMEpi, that potently and selectively inhibits CME" and "CMEpi provides a potent new tool to inhibit CME via a molecularly defined mechanism" are not supported by its data.

I believe we have strong evidence to support the mechanisms by which CMEpi/Wbox2 inhibits endocytosis (through sequestering SNX9 and AP2, see response to referee 3) and hope that our responses to your concerns (above and below) and changes to our manuscript will convince you of the utility of Wbox2 as a novel class of endocytosis inhibitors and the justification for publication in JCB.

JCB policy is that papers are considered through only one major revision cycle. Given the significant remaining reviewer concerns, we unfortunately cannot offer publication of the manuscript.

As argued below, we do not believe the concerns expressed by referee 3 are 'significant' or, in many cases valid. We also do not believe that additional experiments are required; hence we provide a revised manuscript that we believe address your concerns. However, if you disagree and think it is necessary to confirm that the mutant Wbox2 that does not inhibit CME, no longer binds SNX9 or inhibits AP2-TD interactions *in vitro*, we are happy to perform these trivial control experiments.

Response to Reviewer #1 concerns:

The authors have addressed the comments that we raised satisfactorily (with the exception of the below).

Aside from the important concern raised, and addressed below, it appears that this reviewer was satisfied with our responses and revisions and otherwise recommended that the paper be accepted.

I raised the issue below confidentially with the Editor during the first round of review. However, the affected figures remain unchanged in the revised version. I think the journal and authors should address this issue before acceptance.

"We noticed two irregularities in the western blots in this paper and would appreciate it if the journal could use their image integrity checks on them. When we adjusted the contrast on the blots in Fig. 2A and Fig S3A, we noticed that in FigS3A it appears that a band may have been removed and in Fig 2A the blot is presented in a misleading way so as to obscure the leaky expression of TDD (bands can clearly be seen for 5-20 ng/ml after adjusting the contrast). We have not mentioned this in our report."

Obviously, these comments were of great concern to me, given the importance of my (and my postdoc's) reputation for integrity; thus, I quickly checked the figures and the original data. As for Figure 2A, it is true that if the contrast is adjusted so that the other bands are greatly over-exposed one can see very small amounts of TD at higher concentrations of tet (see accompanying figure of the original gel at both contrast settings); these very low levels of 'leaky expression' do not affect rates of Tfn uptake relative to untreated controls. If the journal would like us to also show an overexpressed blot and to mention the very low levels of leaky expression of TD, we are happy to do this.

As for Fig.S3A, again, the original gel is shown here. Endogenous CHC is seen in both lanes and there is no detectable TDD in the lane derived from cells cultured in the presence of high [tet]. The irregularity seen by the reviewer could have arisen when the gel was cropped and rotated, or through digital artefacts as the tif was transferred to ppt (I have observed this in the past).

We ask that these original gels (below) be passed on to this reviewer, regardless of your decision, as it is important that his/her concerns be alleviated.

Response to Reviewer #3

Again, this study is composed of two parallel arms. The first is solid, but rather pedestrian and well studied previously (this is cited in the text).

While we appreciate the reviewer's opinion that the first part of our paper is solid, we disagree that our findings are 'pedestrian and well-studied previously'. To our knowledge no one has studied the effects of TD overexpression on CME and the model that clathrin-EAP interactions

function to drive CCP maturation has not been tested. Indeed, results from Royle and Lemmon have brought this model into question, as clathrin bearing mutations in the TD that disrupt AP2 and EAP binding is able to support CME in yeast (Collette et al., 2009) and mammalian cells (Willox et al., 2012). Moreover, recent genetics studies have identified *de novo* frameshift mutants of CHC that generate C-terminally truncated clathrin mutants corresponding to our TDD construct that are the cause of neurological diseases in humans. We recognize now that we were insufficiently clear in articulating the many novel aspects of our findings, and have significantly revised the Introduction and Discussion to emphasize the significance of our contributions.

It is the second arm (the identification of CHC TD derived cell permeant inhibitors of CME) that would justify acceptance at JCB in my view.

The authors have now responded to the critical issue of whether the isolated peptide pieces of the TD propeller interact directly with AP2 and/or EAPs. Given the measured Kds reported in the all new Figure 9, it appears that the principal concern of all three reviewers has been clearly addressed experimentally.

This reviewer's statement that "*the principal concern of all three reviewers has been clearly addressed experimentally*" to me means that we have indeed responded appropriately to the initial review. If new concerns arise, then we should have a chance to address them.

However, I am unfortunately (still) not convinced.

First there are no negative controls for the ITC experiments, nor peptide alone traces. From the legend, it is impossible to figure out exactly what has been done in panel A. Comparative traces with the mutated (inactivated) CMEpi are necessary.

In conducting these experiments, we followed the advice of Dr. Chad Brautigam, an expert in ITC who runs our Biophysics Core facility. The experiment, which unambiguously shows that the peptide binds to the isolated protein and provides a Kd for this interaction, involves titrating increasing concentrations of peptide, starting with none, into the chamber with the concentrated test protein (either SNX9 or AP2-hinge/ear). 'Peptide only' controls are irrelevant in this experimental as there would be no change in signal with only one entity in solution. Consistent with this, in our literature search of other similar ITC experiments by leading laboratories in our field (see for example Zhuo..Lafer, PMID 25844500, Praefcke...McMahon, PMID 1546985, Mishra...Traub, PMID 1572879) no peptide only controls were performed. We are happy to repeat this experiment with our mutated peptide, which does not inhibit CME, if the editor deems this important.

The same is true for the data presented in panel B of Figure 9. In this pulldown with a highly enriched source of AP2 there is no negative control.

Again, it is unclear what 'negative control' is missing. Following numerous protocols in the literature developed by Kirchhausen, Traub and others, we used GST-TD to pull down AP2 and show that inclusion of our peptide inhibits this interaction in a dose-dependent manner. We have now cited these references and better explained this experiment. The 'negative control' is no peptide.

Also, there is missing the effect of mutating the key residues on the inhibition by the CMEpi peptide.

We are happy to include the mutant peptide, as a specificity control if you would like. However, since this peptide had no effect on endocytosis it is unlikely to bind to SNX9 or inhibit AP2-clathrin interactions. However, as these are new experiments not previously reviewed, we ask permission to address this trivial concern if you see it necessary.

This experiment does not show direct binding of EAPs to either the bound Gst-TD, AP2 or the soluble (but totally unquantified in this work) EAPs in the extract.

The reviewer's statement is somewhat unclear. However, the experiment clearly shows that AP2-TD interactions, whether direct or indirect (although others have clearly established direct interactions) are inhibited by the peptide. This finding confirms our results in vivo (within the complex cytosolic milieu) that the peptide displaces AP2 from CCPs.

What is the conclusion from the right part of panel A and the results on panel B that supposedly explains the incomplete binding of the CMEpi peptide seen in the ITC runs?

Our conclusion is that not all of the isolated AP2 appendage-hinge construct was able to bind peptide suggesting that this construct might not all be properly folded and/or that the unstructured hinge precludes peptide binding in some conformations, which we have now more clearly stated in the text. Incomplete binding to recombinant proteins is not uncommon. Hence we used a second method to directly assess the ability of the peptide to interfere with native AP2-TD interactions. I would argue that this shows rigor in characterizing the mechanism of inhibition.

Overall, Panel B presents a complicated experiment and the side by side use of a strong TD-ear and a TD--EAP inhibitor should be considered for comparison.

We respectfully disagree and believe that this simple experiment most closely represents the ability of the peptide to perturb AP2-clathrin interactions in intact cells. We are unaware of a 'strong TD-Ear and TD-EAP inhibitor'.

The absence of these essential controls makes the results open to question. In this reviewer's mind, the matter is critical given the mechanistic claims and novelty prominence of the inhibitory very short TD peptides in this manuscript.

As argued above, we disagree that 'essential controls' are absent and provide, but two well-established methods that Wbox2 directly binds with AP2 and SNX9 and disrupts AP2-TD interactions.

Second, because the Kds are in the range of typical EAP interactions, an attempt to decipher whether the inhibitory sequences from the TD have any meaningful similarity to known EAP binding sequences (eg. role/positioning of hydrophobic residues) should be undertaken.

This statement does not make sense to us. The inhibitory peptide mimics contact sites on the TD, not EAP binding sequences. Moreover, we show that when the residues on the peptide corresponding to critical residues on the TD are mutated, the activity of the peptide is lost.

April 15, 2020

RE: JCB Manuscript #201908189RR-A

Dr. Sandra L. Schmid
UT Southwestern Medical Center
6000 Harry Hines Blvd
Dallas, Texas 75390

Dear Dr. Schmid:

Thank you for submitting your revised manuscript entitled "Structure-based inhibitors reveal roles for the clathrin terminal domain and its W-box binding site in CME". The JCB editorial team has discussed the paper and reviewer comments in depth and we would be happy to publish your paper pending changes to the text to tone down some of the structural biology and mechanistic claims (please see marked up pdf) and final revisions necessary to meet our formatting guidelines (see details below).

- Provide the main and supplementary texts as separate, editable .doc or .docx files
- Provide main and supplementary figures as separate, editable files according to the instructions for authors on JCB's website *paying particular attention to the guidelines for preparing images and blots at sufficient resolution for screening and production*
- Format references for JCB
- Add MW markers to blots in Figure 4B
- Add scale bars to figure S1C inset, S3C and inset, S3E inset, S4
- Add paragraph after the Materials and Methods section briefly summarizing all "Online Supplementary Materials" - incorporate figure titles, supplemental tables and videos in this section.

A. MANUSCRIPT ORGANIZATION AND FORMATTING:

Full guidelines are available on our Instructions for Authors page, <http://jcb.rupress.org/submission-guidelines#revised>. **Submission of a paper that does not conform to JCB guidelines will delay the acceptance of your manuscript.**

B. FINAL FILES:

-- High-resolution figure and video files: See our detailed guidelines for preparing your production-ready images, <http://jcb.rupress.org/fig-vid-guidelines>.

Thank you for this interesting contribution, we look forward to publishing your paper in Journal of Cell Biology.

Sincerely,

Jodi Nunnari, Ph.D.
Editor-in-Chief

Marie Anne O'Donnell, Ph.D.
Scientific Editor

Journal of Cell Biology